# D$^2$GS: Dense Depth Regularization for LiDAR-free Urban Scene Reconstruction

**Kejing Xia**[1]* **Jidong Jia**[2]* **Ke Jin**[3] **Yucai Bai**[4] **Li Sun**[4] **Dacheng Tao**[5] **Youjian Zhang**[4]†

[1]Wuhan University  [2]Shanghai Jiaotong University  [3]TongJi University  [4]Bosch
[5]Nanyang Technological University

xiakejing.lesia@whu.edu.cn, jjd1123@sjtu.edu.cn, 2330379@tongji.edu.cn
dacheng.tao@ntu.edu.sg, {Yucai.BAI, Kevin.SUN, Youjian.ZHANG}@cn.bosch.com

## Abstract

Recently, Gaussian Splatting (GS) has shown great potential for urban scene reconstruction in the field of autonomous driving. However, current urban scene reconstruction methods often depend on multimodal sensors as inputs, *i.e.* LiDAR and images. Though the geometry prior provided by LiDAR point clouds can largely mitigate ill-posedness in reconstruction, acquiring such accurate LiDAR data is still challenging in practice: i) precise spatiotemporal calibration between LiDAR and other sensors is required, as they may not capture data simultaneously; ii) reprojection errors arise from spatial misalignment when LiDAR and cameras are mounted at different locations. To avoid the difficulty of acquiring accurate LiDAR depth, we propose D$^2$GS, a LiDAR-free urban scene reconstruction framework. In this work, we obtain geometry priors that are as effective as LiDAR while being denser and more accurate. **First**, we initialize a dense point cloud by back-projecting multi-view metric depth predictions. This point cloud is then optimized by a Progressive Pruning strategy to improve the global consistency. **Second**, we jointly refine Gaussian geometry and predicted dense metric depth via a Depth Enhancer. Specifically, we leverage diffusion priors from a depth foundation model to enhance the depth maps rendered by Gaussians. In turn, the enhanced depths provide stronger geometric constraints during Gaussian training. **Finally**, we improve the accuracy of ground geometry by constraining the shape and normal attributes of Gaussians within road regions. Extensive experiments on the Waymo dataset demonstrate that our method consistently outperforms state-of-the-art methods, producing more accurate geometry even when compared with those using ground-truth LiDAR data.

## 1  Introduction

Modeling urban street scenes is fundamental to autonomous driving [2]. Accurate 3D reconstructions of roads, buildings, and street furniture enable the creation of high-fidelity virtual environments, which are crucial for the closed-loop simulation and testing of perception, planning, and control systems [3]. The ability to reuse and edit these digital urban scenes, rather than repeatedly collecting fresh data for every test scenario, significantly improves the efficiency of data collection.

Recently, methods leveraging Neural Radiance Field (NeRF) [4] and 3D Gaussian Splatting (3DGS) [5] have become popular in this field for their photorealistic results. However, NeRF-based methods are computationally intensive while 3DGS is favored due to its explicit representation, high

---

*This work was completed during an internship at Bosch. Equal contribution.
†Corresponding author.

39th Conference on Neural Information Processing Systems (NeurIPS 2025).

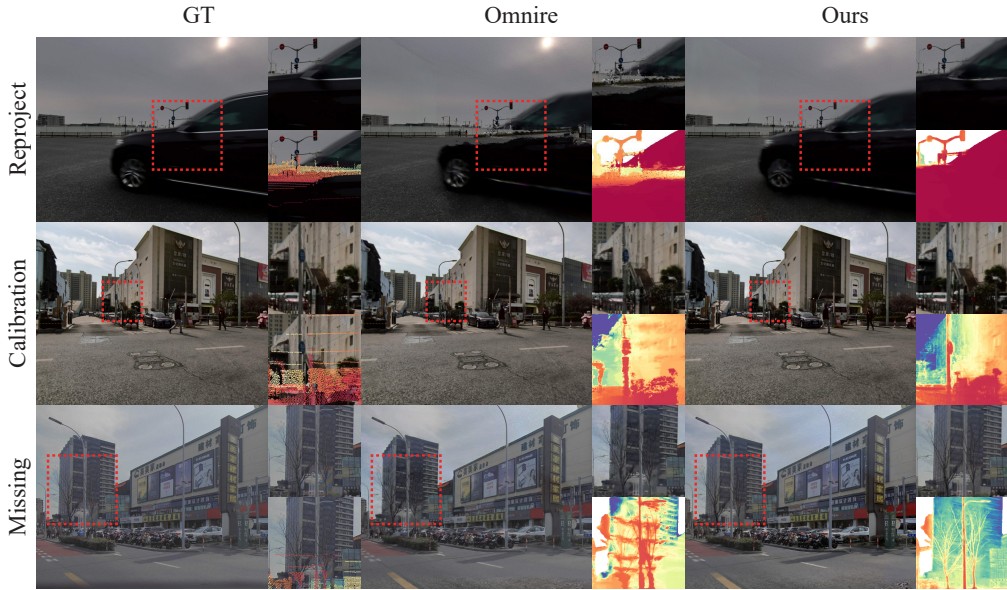

Figure 1: Examples of common LiDAR acquisition issues: a) reprojection error, b) calibration misalignments, and c) LiDAR missing problem. OmniRe [1] yields poor performance in both image reconstruction and depth estimation when using these inaccurate LiDAR measurements.

efficiency, and rapid rendering speed [6]. In fact, 3DGS has demonstrated its powerful reconstruction ability for dynamic urban scenes [7, 8, 1, 9, 10, 11].

Most of the 3DGS urban street scene reconstruction methods rely on LiDAR data to provide an accurate geometry prior. Usually, this prior can be applied in two aspects: i) LiDAR point cloud can be utilized to initialize Gaussian points, ii) LiDAR point cloud can provide a sparse depth supervision during the training phase. However, there are several notable drawbacks when acquiring LiDAR data in practice. First, the acquisition of LiDAR data requires specialized vehicles and costly equipment [12], which makes the data collection process challenging and less scalable. Second, the calibration between sensors (LiDARs and cameras) is highly demanding. Since these sensors do not capture data simultaneously, both spatial and temporal calibration are required to accurately align the LiDAR points with images. Furthermore, in practical data collection systems, LiDAR sensors and cameras are inevitably mounted at different positions and heights, causing significant viewpoint disparities and resulting in reprojection errors when LiDAR points are projected onto images (Fig. 1). However, most existing methods use this LiDAR projection result as ground-truth depth map. In preliminary experiments, we found that misalignment between LiDAR projections and pixels significantly affects the reconstruction results (as shown in Fig. 1). Finally, LiDAR provides only sparse depth supervision during the GS training phase, limiting the model's ability to learn fine-grained geometric details.

In the meantime, many recent studies [13, 14, 15, 16, 17, 18, 19, 20, 21] have also explored ways to incorporate geometric constraints into the optimization of sparse-view scene reconstruction. However, these methods either rely on monocular depth estimation [13, 14, 15, 16, 17, 18], which only provides relative depth, or obtain metric depth through multi-view depth estimation [19, 20, 21], which is unsuitable for dynamic scenes and still suffers from large absolute errors.

To overcome the challenges of acquiring accurate geometry prior using LiDAR points, while obtaining a more robust and dense depth for regularization, we introduce $D^2GS$ – a LiDAR-free dynamic urban street scene reconstruction framework. First, to obtain an initial point cloud as a replacement for LiDAR point clouds, we leverage depth predictions from a pre-trained multi-view estimation model [21]. Re-projecting these depths yields an initial, dense point cloud. However, this point cloud is per-pixel, resulting in an excessive number of points. To manage the computational cost, we introduce a progressive pruning strategy. This strategy starts training the Gaussian field with minimal scene graph components, then gradually prunes the less important Gaussians to retain a compact yet representative point set that effectively captures the global scene geometry. Second, for the absence of

depth supervision provided by LiDAR data, we propose a novel joint optimization strategy for depth and the Gaussian representation. Inspired by incorporating image-based diffusion priors [22] used in Gaussian training, we introduce a Depth Enhancer module that leverages priors from the depth foundation model [23] to refine the depth rendered from the current Gaussian state. Specifically, we iteratively refine and update the densely rendered depth by using the current geometric information as guidance within the diffusion framework, and using refined depth for Gaussian training. The refined depth maps can offer a dense and accurate depth supervision that is semantically consistent with the scene, making them a superior alternative to LiDAR point clouds. Finally, since the ground plane possesses strong geometric priors in urban scenes [10], we initialize and optimize a dedicated road node in the scene graph for a better geometry and depth estimation of the road. Specifically, we constrain each Gaussian in the road node to be a planar parallel to a precomputed ground plane by regularizing their shape and normal attributes. In summary, our main contributions are as follows:

- We introduce $D^2GS$, a framework for reconstructing dynamic urban street scenes using only camera inputs, eliminating the need for LiDAR points acquisition and the error brought by calibration and reprojection.

- We propose a progressive pruning strategy to efficiently manage the dense point cloud derived from initial depth estimates, yielding a compact Gaussian representation that captures global scene geometry.

- We develop a joint optimization strategy using a diffusion-based Depth Enhancer that iteratively refines estimated depths and the Gaussian representation, providing robust geometric supervision with dense metric depths.

- We further enhance the reconstruction accuracy on roads by creating a Road Node in the scene graph, explicitly modeling the ground plane using strong geometric priors.

## 2 Related Works

**Urban scene reconstruction.** Neural Radiance Field (NeRF) [4] and 3D Gaussian Splatting (3DGS) [5] have recently become the leading solutions for urban scene reconstruction. Early methods reconstruct entire city blocks and improve scalability to larger environments [24, 25], but all assumed static scenes. To handle dynamic elements, methods such as SUDS [26] and EmerN-eRF [27] employ self-supervised decompositions into static and dynamic components, while neural scene-graph approaches explicitly disentangle motion and structure [28] and then leverage instance-aware frameworks to decompose each instance and model each moving object independently [29]. Nowadays, S3GS [7] introduces deformable fields for Gaussians to capture scene dynamics, and PVG [8] assigns each Gaussian a lifespan attribute for temporal modeling. Then scene graphs are also used in GS-based method, StreetGaussians [9] represents static and dynamic components as individual Gaussians within a Gaussian scene graph and optimizes them jointly, and OmniRe [1] integrates multiple Gaussian types to better reconstruct dynamic elements.

**Depths in 3D Gaussian.** Depth information provides essential geometric priors for resolving ambiguities in 3D reconstruction [19] and has been widely used in sparse-view reconstruction fields. In scenarios lacking ground-truth depth, many works adopt monocular depth estimates as weak supervision, leveraging scale-invariant loss functions [13, 14, 15] and depth ranking constraints [16, 17]. Recent advances such as DNGaussian [18] further improve this paradigm with patch-wise depth normalization. However, these monocular supervision methods inherently suffer from scale ambiguities due to the unknown shift and scale factors in predicted depth [21]. While some feed-forward 3DGS methods like MVSplat [20] and DepthSplat [21] address this by incorporating multi-view depth estimation, but they cannot handle dynamic scenes, and feed-forward depths are imperfect for further training. In contrast, our method leverages metric depth supervision while progressively refining depth predictions with generative priors through an iterative training scheme.

**Diffusion priors in 3D Gaussian.** Generative priors have recently been incorporated into 3DGS to reduce artifacts arising from geometric inaccuracies in novel-view synthesis [30, 22]. These approaches leverage diffusion models to enforce semantic coherence, hallucinate photometric details, and extrapolate occluded regions. DriveDreamer4D [31] further adapts this framework to dynamic driving scenarios. However, all of these methods concentrate on image-space priors and overlook the orthogonal role of depth-space generative priors. Diffusion-based monocular depth estimators such as Marigold [32] establish a foundation for learned depth priors, and Marigold-DC [23] shows

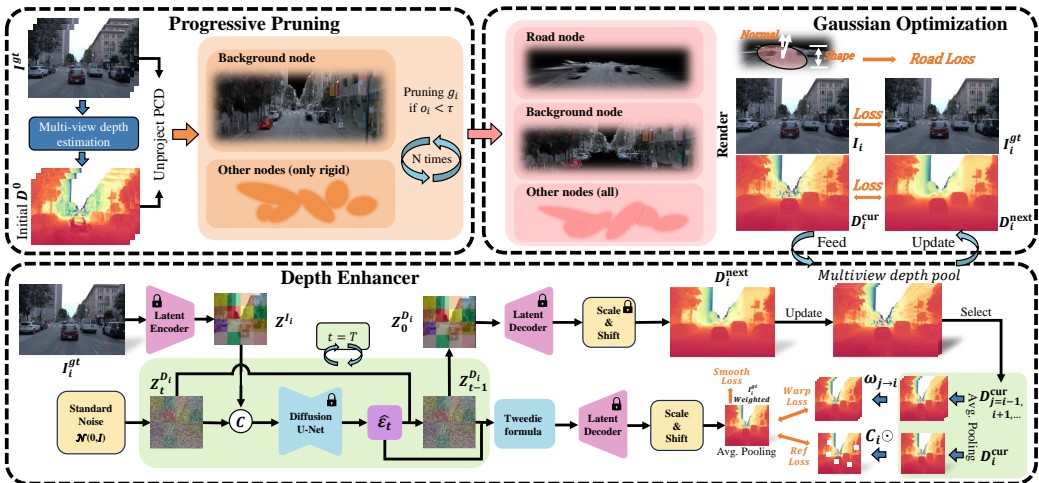

Figure 2: Pipeline of D²GS. We first employ a Progressive Pruning strategy to obtain a robust global Gaussian initialization. A Road Node is incorporated into the scene graph structure to regularize the road region using strong geometric priors. During training, Gaussian optimization and depth refinement are performed iteratively, allowing depth to be learned jointly from Gaussian supervision and enhanced by diffusion priors from a pretrained depth foundation model.

that classifier-guided sampling can propagate sparse metric depths into dense predictions. However, our approach diverges by directly integrating Marigold's generative depth prior into the 3DGS optimization loop to refine the dense imperfect depths.

## 3 Method

Our proposed D²GS (Fig. 2) addresses the challenge of LiDAR-free dynamic urban scene reconstruction through three key components. First, we propose an initialization strategy that leverages estimated depth as input and applies progressive pruning to obtain a compact Gaussian representation. (Sec. 3.1). Second, during Gaussian optimization, we propose a multi-view Depth Enhancer module to iteratively refine the rendering depth, providing dense geometric supervision (Sec. 3.2). Third, we incorporate a specialized road node into the scene graph representation to explicitly model the ground plane using strong geometric priors (Sec. 3.3). We illustrate these components in the following subsections.

### 3.1 Initialization via Progressive Pruning

Effective initialization of the 3D Gaussians is crucial for achieving high-quality reconstruction results with Gaussian Splatting. While most current methods [9, 1] rely on LiDAR point clouds for this purpose, our LiDAR-free setting requires an alternative solution. We begin by predicting an initial depth map for each input image using a pre-trained multi-view depth estimation network [21], denoted as $D_i$ for $i_{th}$ image. Given the camera pose and $D_i$, we unproject each depth map into a point cloud. By aggregating these point clouds, we obtain a unified point cloud $\mathbf{P}$.

The point cloud $\mathbf{P}$ contains as many points as image pixels, resulting in an excessively large size (about 100 million) that is computationally impractical for Gaussian training. To address this challenge, one common solution is to de-densify the points by voxel grids. However, this can lead to the loss of fine geometric details. To mitigate this, we propose a progressive pruning strategy that learns to de-densify the points in a learnable manner. Specifically, we initialize Gaussians with a non-deformable and low-resolution training setting to optimize these points and progressively eliminate less important Gaussians based on their opacity, *i.e.* the significance of the Gaussian in the scene representation. After each round of progressive pruning, we reset the opacity values and continue the next round. The progressive pruning process can be formulated as:

$$\mathbf{G}^{(\mathrm{N}+1)} = \left\{ g_i \in \mathbf{G}^{(\mathrm{N})} \,\middle|\, o_i^{(\mathrm{N})} \geq \tau \right\},$$

where $\mathbf{G}^{(\mathrm{N})}$ represents the set of Gaussians before pruning at round $N$, $o_i^{(\mathrm{N})}$ is the opacity of one Gaussian point $g_i$, and $\tau$ is the opacity threshold. This progressive strategy produces a manageable set of Gaussians that capture the essential scene geometry, effectively balancing computational efficiency and the quality of the initial geometric representation.

## 3.2 Joint Optimization of Depth and Gaussian Splatting

In the absence of LiDAR depth, obtaining reliable metric depth as a geometric constraint becomes challenging. Our experiments found that the estimated metric depths [21] are not sufficiently accurate for training, as directly applying these noisy depth maps as supervision leads to significant degradation in both reconstruction quality and geometric accuracy. (refer to OmniRe+DS results in Tab. 2).

On the one hand, Gaussian training can infer depth leveraging spatial consistency, and correct inconsistent depth estimates [21]. On the other hand, metric depth estimation, which aligns closely with image semantics, can provide smooth and fine-grained depth supervision, especially when training views are insufficient. To marry the advantages of both sides, we propose a joint optimization strategy that iteratively refines both the depth predictions and the Gaussian parameters. We perform iterative depth optimization as part of the Gaussian optimization process. For each depth optimization iteration, we first identify the high-confidence region from the previously estimated depth map, then refine the dense depth with the geometric information guided by high-confidence regions, using current Gaussian geometry to condition the *Depth Enhancer*. We then update the supervision (Multiview) depth pool to guide the Gaussian learning process.

**Identify the High-confidence Region.** Inspired by methods handling uncertain inputs [33, 34], we introduce a confidence measure based on the consistency between the previously estimated depth map and the currently rendered depth. The high-confidence region reflects the consistency between Gaussian training and depth estimation, indicating high accuracy. Specifically, for each pixel in view $i$, we compute the relative difference between the current depth $\mathbf{D}_i^{\mathrm{cur}}$ and the previous iteration's depth $\mathbf{D}_i^{\mathrm{prev}}$. The confidence map $\mathbf{C}_i$ is formulated as:

$$\mathbf{C}_i \;=\; \mathbb{I}\Big\{ \max\!\big( \mathbf{D}_i^{\mathrm{cur}} \oslash \mathbf{D}_i^{\mathrm{prev}}, \; \mathbf{D}_i^{\mathrm{prev}} \oslash \mathbf{D}_i^{\mathrm{cur}} \big) \;<\; \lambda_C \Big\}, \tag{1}$$

where $\oslash$ denotes element-wise division, $\mathbb{I}$ is an element-wise indicator function that evaluates to 1 if the enclosed condition is true, and 0 otherwise.

**Multi-view Depth Enhancer for Depth Refinement.** We employ a Multi-view Depth Enhancer module to refine the depth map, especially the low-confidence region. Inspired by the Marigold-DC [23], the proposed module leverages generative priors from the pre-trained monocular depth prediction model, *i.e.*, Marigold [32], by optimizing latent depth representations and scale-shift parameters through classifier-free guidance. However, it differs from Marigold-DC in the guidance accuracy: instead of relying on sparse ground-truth depth, our Depth Enhancer's diffusion steps are guided by the potentially noisy depth estimates $\mathbf{D}_i$ which are rendered by the Gaussian at the current state. Therefore, we design three losses to suppress the potential noise and enforce multi-view geometric consistency to obtain more accurate estimates.

**i) Reference loss $L_{\mathbf{ref}}$.** Similar to Marigold-DC that use sparse ground-truth depth to inject metric depth information, we also use the rendered depth as a reference signal. However, we do not use it directly as supervision since the rendered depth $\mathbf{D}_i$ is not reliable. Instead, we first downsample both the predicted depth $\hat{\mathbf{D}}_i^{(k)}$ (the predicted depth map for view $i$ at diffusion step $k$) and the reference depth $\mathbf{D}_i$ to a patch level (size $p \times p$ with average pooling, denoted by $\Downarrow$). This reduces the influence of local fluctuations. The loss then computes a combination of L1 and L2 distances between the downsampled maps, modulated by the confidence map $\mathbf{C}_i$ from Eq. 1:

$$L_{\mathrm{ref}} = \Downarrow \mathbf{C}_i \odot \left( \left\| \Downarrow \hat{\mathbf{D}}_i^{(k)} - \Downarrow \mathbf{D}_i \right\|_1 + \left\| \Downarrow \hat{\mathbf{D}}_i^{(k)} - \Downarrow \mathbf{D}_i \right\|_2^2 \right). \tag{2}$$

The confidence-weighted loss ensures that the refinement closely follows the reference depth in high-confidence regions, while in uncertain areas, it guides the generation of metric depth based on both the reference depth and the image semantics.

**ii) Multi-view warping loss** $L_w$. It is also essential to ensure that the depth remains consistent across multiple viewpoints. Thus, we employ a projection mechanism to further guide the depth generation. Specifically, the depth warping operation $\mathcal{W}$ transforms the predicted depth map from another view $j$ to the view of the reference frame $i$ using the given camera poses. Then we penalize cross-view inconsistency via an L1 loss between warped and predicted depths:

$$L_w = \sum_{j \neq i} \left\| \mathcal{W}_{j \to i}(\mathbf{D}_j) - \hat{\mathbf{D}}_i^{(k)} \right\|_1, \tag{3}$$

The gradient of the warping loss enforces the geometry consistency between the predicted depth and the depth of the adjacent frames.

**iii) Smooth loss** $L_{\text{smooth}}$. To promote smoothness in the refined depth map, we incorporate an edge-aware smoothness constraint on the predicted depth $\hat{\mathbf{D}}_i^{(k)}$, conditioned on the ground-truth input image $\mathbf{I}_i^{gt}$:

$$L_{\text{smooth}} = \frac{1}{HW} \sum_{x,y} \left( \left| \partial_x \hat{\mathbf{D}}_i^{(k)}(x,y) \right| e^{-|\partial_x \mathbf{I}_i(x,y)|} + \left| \partial_y \hat{\mathbf{D}}_i^{(k)}(x,y) \right| e^{-|\partial_y \mathbf{I}_i(x,y)|} \right), \tag{4}$$

where $\partial_x$ and $\partial_y$ denote the spatial gradients in the horizontal and vertical directions, respectively. The weight $e^{-|\partial \mathbf{I}_i(x,y)|}$ reduces the influence of the smooth loss in areas with strong edge.

Finally, the total loss function guiding the diffusion optimization at each step is a weighted sum:

$$L_{\text{diffuse}} = \lambda_{\text{ref}} L_{\text{ref}} + \lambda_{\text{smooth}} L_{\text{smooth}} + \lambda_w L_w. \tag{5}$$

During the Gaussian training phase, the Depth Enhancer iteratively refines the depth $\mathbf{D}_i^{\text{cur}}$ to $\mathbf{D}_i^{\text{next}}$ every $N$ iterations in order to balance time costs and effectiveness. More details of the iterative training scheme will be illustrated in the supplementary.

## 3.3 Optimization of Road Node in Gaussian Scene Graph

Roads are a key component of urban scenes and exhibit strong geometric priors, *i.e.* a specific height that can be precomputed from the point cloud and a known planar structure. Similar to [35, 10, 36], we also take advantage of the prior knowledge of roads to obtain a better depth/geometry on road region. Specifically, we introduce a specialized Road Node within our Gaussian scene graph structure, initializing it with all points identified as belonging to the road through segmentation. Then two geometry regularization are applied:

**Position Constraint.** We pre-compute the average height of the initial road point cloud and constrain the positions of the road Gaussians to stay close to this reference height.

$$L_{\text{plane}} = \frac{1}{|\mathbf{G}_{\text{road}}|} \sum_{g_i \in \mathbf{G}_{\text{road}}} \left( \mu_{z,i} - \bar{\mu}_z \right)^2, \tag{6}$$

where $G_{\text{road}}$ is the set of Gaussians assigned to the Road Node, $\mu_{z,i}$ denotes the z-coordinate of the center of Gaussian $g_i$, and $\bar{\mu}_z$ is the mean road height of the initial points on the road.

**Planar Constraint.** Also, the shape and orientation of individual Gaussians within the Road Node should be constrained as a flat planar. Specifically, we encourage the Gaussian to be flat along the ground. Similar to [36], this constraint is formulated as,

$$L_{\text{shape}} = \frac{1}{|\mathbf{G}_{\text{road}}|} \sum_{g_i \in \mathbf{G}_{\text{road}}} \left( \lambda_{\text{normal}} \| \mathbf{n}_i - \hat{\mathbf{z}} \|_2 + \lambda_{\text{flat}} \| s_{\min,i} \|_1 \right), \tag{7}$$

where $\lambda_{\text{normal}}$ and $\lambda_{\text{flat}}$ are hyperparameters. For each Gaussian, $s_{\min,i}$ denote the smallest scale and the normal $\mathbf{n}_i$ derives from the the axis aligns with smallest scale. $\hat{\mathbf{z}}$ is the normal of the ground. The overall loss for the Road Node is the sum of these two components: $L_{\text{road}} = L_{\text{plane}} + L_{\text{shape}}$.

## 3.4 Loss functions for Gaussian Training

In Gaussian training iterations, we follow the training loss of OmniRe [1], except for the road loss:

$$L = (1 - \lambda_r) L_1 + \lambda_r L_{\text{SSIM}} + \lambda_{\text{road}} L_{\text{road}} + \lambda_{\text{depth}} L_{\text{depth}} + \lambda_{\text{opacity}} L_{\text{opacity}} + L_{\text{reg}}, \tag{8}$$

Table 1: Comparison of image reconstruction performances on Waymo NOTR Dynamic32 Dataset. OmniRe+DS refers to OmniRe using DepthSplat's estimated depth for depth regularization.

| Inputs | Methods | Image reconstruction | | | Novel view synthesis | | |
| --- | --- | --- | --- | --- | --- | --- | --- |
| | | PSNR ↑ | SSIM ↑ | LPIPS ↓ | PSNR ↑ | SSIM ↑ | LPIPS ↓ |
| Img+LiDAR | EmerNeRF | 28.16 | 0.806 | 0.228 | 25.14 | 0.747 | 0.313 |
| | 3DGS | 28.47 | 0.876 | 0.136 | 25.14 | 0.813 | 0.165 |
| | S3GS | 31.35 | 0.911 | 0.106 | 27.44 | 0.857 | 0.137 |
| | PVG | 34.25 | 0.936 | 0.111 | 28.83 | 0.855 | 0.159 |
| | OmniRe | 33.35 | 0.945 | 0.067 | 29.02 | 0.871 | 0.115 |
| Img | OmniRe w/o LiDAR | 34.07 | 0.947 | 0.079 | 27.69 | 0.853 | 0.146 |
| | OmniRe+DS | 31.26 | 0.932 | 0.089 | 27.37 | 0.854 | 0.136 |
| | Ours | 34.34 | 0.950 | 0.074 | 28.80 | 0.867 | 0.129 |

Table 2: Comparison of depth estimation performances among multiple LiDAR-free methods and a multi-view depth estimation method DepthSplat on Waymo NOTR Dynamic32 Dataset. All metrics are computed with sparse GT depth from LiDAR.

| Inputs | Methods | L1 ↓ | Abs. Rel. ↓ | RMSE ↓ | $\delta < 1.25$ ↑ |
| --- | --- | --- | --- | --- | --- |
| Img | DepthSplat | 8.7088 | 0.5412 | 14.6494 | 0.6020 |
| | OmniRe w/o LiDAR | 4.2197 | 0.1977 | 8.3008 | 0.7212 |
| | OmniRe+DS | 5.5345 | 0.3137 | 8.5426 | 0.6422 |
| | Ours | 2.6043 | 0.1313 | 4.6723 | 0.8404 |

where $L_1$ and $L_{SSIM}$ are the L1 and SSIM losses on rendered images, $L_{depth}$ compares the rendered depth of Gaussians with dense reference depth, $L_{opacity}$ encourages the opacity of the Gaussians to align with the non-sky mask, and $L_{reg}$ represents various regularization terms applied to different Gaussian representations.

## 4 Experiments

### 4.1 Experimental Setup

**Datasets.** We conducted a detailed experimental analysis using the Waymo dataset [37], a widely-used dataset for real-world driving scenarios. To evaluate our method's performance in dynamic urban environments, we selected the challenging Waymo dataset subset, NOTR Dynamic32, proposed by [27]. We utilize data from the three frontal cameras (FRONT, FRONT RIGHT, FRONT LEFT), resizing the images to a resolution of $640 \times 960$.

**Metrics.** We adopt PSNR, SSIM, and LPIPS as metrics to evaluate the quality of image reconstruction. Since our method does not use ground-truth point cloud depth for supervision, we employ L1, RMSE, Abs.Rel., and Threshold accuracy($\delta$) [21] to assess the depth estimation performance. Specifically, these metrics are evaluated with the sparse ground-truth depth obtained from LiDAR.

**Baselines.** We compare our method against several Gaussian Splatting approaches: 3DGS [5], S3GS [7], PVG [8], and OmniRe [1], and NeRF-based approach EmerNeRF [27]. However, these methods all require a LiDAR point cloud for optimization. For a fairer comparison, we include two additional baselines, (1) **OmniRe w/o LiDAR**, which trains OmniRe without any depth regularization, and (2) **OmniRe + DS**, which trains OmniRe using depth estimates from DepthSplat [21] in place of LiDAR depths. For more implementation details, please refer to the supplemental material.

### 4.2 Evaluation on Urban Driving Dataset

#### 4.2.1 Image Reconstruction Performance

Quantitative results for image reconstruction and novel view synthesis, presented in Tab. 1, demonstrate that our $D^2GS$ method consistently outperforms all other methods and is even comparable to methods that use ground-truth (GT) LiDAR. Notably, OmniRe [1] without depth supervision shows competitive image quality on training views. We infer that the absence of explicit geometric

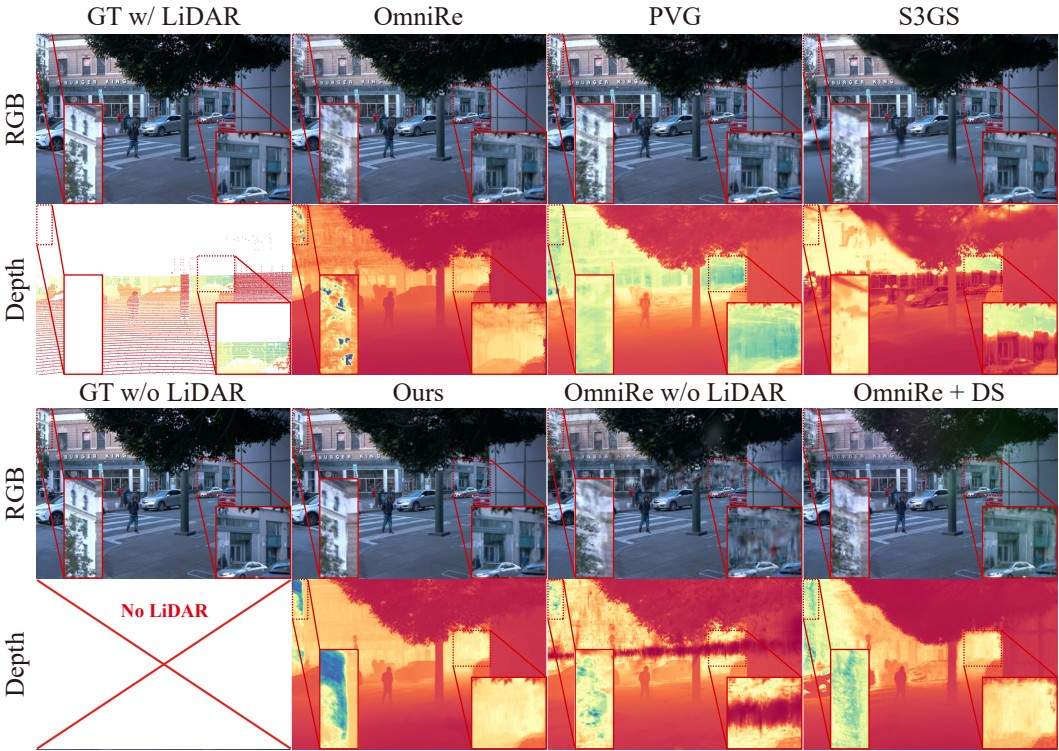

Figure 3: Comparison of image reconstruction and depth estimation performances on Waymo NOTR Dynamic32 Dataset with S3GS [7], PVG [8], OmniRe [1], and LiDAR-free baselines. Zoom in for better visual comparison.

constraints allows its Gaussian representation to overfit to the training views, resulting in poor generalization to novel views (PSNR drops from 34.07 to 27.69). The results of OmniRe+DS also validates that using the inaccurate depth prediction as supervision will in turn degrade reconstruction quality.

As shown in Fig. 3, $D^2GS$ renders more coherent images with superior reconstruction of fine details like vehicle contours and building textures. In contrast, former methods with LiDAR supervision may suffer from a quality degradation where the LiDAR is missing: the results of OmniRe [1] PVG [8] and S3GS [7] become blurry in the upper region. We can also observe the inaccurate depth rendering in the corresponding regions, S3GS [7] shows an evident dividing line in the depth map. As for LiDAR-free methods, although Gaussian training can overfit to images in the training views, it suffers from distortion and floater artifacts in novel views. Visualizations of the corresponding depth maps further confirm that it is difficult to recover accurate depth either by relying solely on Gaussian training (OmniRe w/o LiDAR) or on noisy depth predictions (OmniRe + DS).

### 4.2.2 Depth Estimation Performance

As shown in Tab. 2, $D^2GS$ achieves significant improvement on depth estimation accuracy compared to methods without GT LiDAR supervision. First, DepthSplat, which provides our initial depth estimates, shows inferior performance. This may be due to its inability to handle dynamic scenes. Moreover, compared to DepthSplat [21], the OmniRe + DS method shows an improvement, suggesting that Gaussian training has some capacity to correct noisy depth predictions. However, it still performs worse than OmniRe w/o LiDAR. We infer that, while depth supervision can serve as an effective regularizer for Gaussian training, inaccurate depth predictions may mislead the optimization process. In contrast, the depth accuracy of our method achieves approximately a 53% improvement over OmniRe + DS, validating the effectiveness of our proposed modules, given that the same initialization is employed.

A similar conclusion can be drawn from the Qualitative results (Fig. 3), as we have provided some analysis in Sec. 4.2.1. Our proposed method generates accurate dense depth maps, which offer

Table 3: Ablation study of our proposed key components and training strategy.

| Components | | | Settings of D.E. | | | | | | | |
|---|---|---|---|---|---|---|---|---|---|---|
| R.N. | P.P. | D.E. | $L_{ref}$ | $L_w$ | $L_{smooth}$ | Update | L1 ↓ | Abs. Rel. ↓ | RMSE ↓ | $\delta < 1.25$ ↑ |
| × | ✓ | ✓ | ✓ | ✓ | ✓ | Real-time | 2.7580 | 0.1535 | 4.7748 | 0.8080 |
| ✓ | × | ✓ | ✓ | ✓ | ✓ | Real-time | 2.9121 | 0.1491 | 5.1675 | 0.8041 |
| ✓ | ✓ | × | × | × | × | Real-time | 2.8190 | 0.1595 | 5.3086 | 0.8107 |
| ✓ | ✓ | ✓ | ✓ | × | × | Real-time | 2.6477 | 0.1488 | 4.6977 | 0.8163 |
| ✓ | ✓ | ✓ | ✓ | ✓ | ✓ | Periodic | 2.9168 | 0.1700 | 5.0385 | 0.7701 |
| ✓ | ✓ | ✓ | ✓ | ✓ | ✓ | Real-time | **2.3847** | **0.1303** | **4.2854** | **0.8526** |

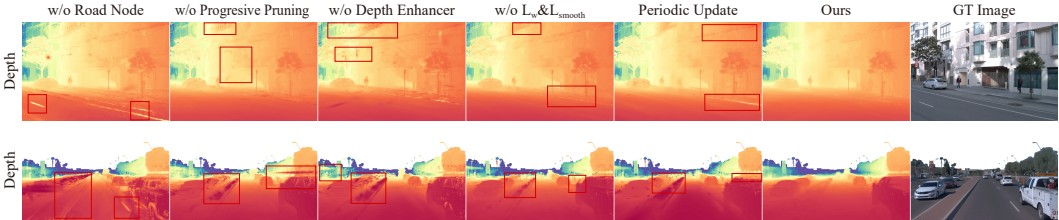

Figure 4: Visualization comparison of the proposed modules and training strategy. Zoom in for better comparison.

effective regularization and suppress floaters in regions where the depth data is missing or unreliable. It is worth noting that, apart from the issue of missing upper LiDAR points, the Waymo dataset exhibits minimal reprojection and calibration errors, as it is a meticulously calibrated dataset. However, such errors are common in real-world, industrial-scale data, where our method will demonstrate more significant improvements. More visual comparisons are provided in Supplementary.

## 4.3 Ablation Studies

Building upon the previous results, we perform ablation studies on a Waymo subset (12 sequences, detailed in supplementary material) to assess the impact of $D^2GS$'s key components: Progressive Pruning (P.P.), Depth Enhancer (D.E.), and Road Node (R.N.). We evaluate variants by removing each component individually. We further analyze the loss function and update strategy of the Depth Enhancer.

### 4.3.1 Network Components

**Effectiveness of Progressive Pruning.** Without progressive pruning ($D^2GS$ w/o P.P.), Gaussians are initialized directly from a dense point cloud. The results (Tab. 3) are worse because initial noise and outliers in the dense points lead to flawed Gaussian representations. Floaters arise when training from this setting (Fig. 4). Our P.P. strategy successfully filters these initial points, creating a robust global geometry for the following reconstruction.

**Effectiveness of Depth Enhancer.** Removing the Depth Enhancer ($D^2GS$ w/o D.E.) degrades depth estimation accuracy and quality (Tab. 3). The D.E. iteratively refines depth rendered from current Gaussians using diffusion priors, establishing a crucial loop for geometric accuracy. Without it, the model struggles with insufficient geometric constraints, failing to capture fine geometric details and exhibiting depth errors and distortions.

**Effectiveness of Road Node Constraint.** Excluding the Road Node constraint ($D^2GS$ w/o R.N.) removes the strong ground plane prior, leading to a decline in depth metrics (Tab. 3) and qualitative artifacts such as uneven ground surfaces (Fig. 4). Inferring geometry for large, textureless surfaces like roads from multi-view images alone is challenging. Road Nodes enforce shape and normal constraints on Gaussians in detected road areas, promoting smooth and continuous ground geometry.

#### 4.3.2 Settings of Depth Enhancer

**Loss Function.** We ablate the D.E.'s loss function by removing its projection and smooth loss terms. As shown in Tab. 3, this degrades D.E. performance and overall reconstruction quality. These loss terms provide geometric information from the Gaussian with less error, enabling the D.E. to generate more consistent and accurate depth maps, which in turn provide more effective geometric supervision during iterative optimization, validating our design.

**Update Strategy.** We compare our adopted *Real-time Update* strategy for the D.E. with a *Periodic Update* strategy. In *Periodic Update*, depth maps for all training views are updated at the same iteration. In our *Real-time Update*, only a single depth map of one training view is rendered, enhanced, and used for depth regularization during a depth enhancement iteration. Experimental results (Tab. 3) show that *Real-time Update* significantly outperforms *Periodic Update*. The reason is that *Periodic Update* suffers from using outdated depth for supervision, which leads to a conflict between Gaussian optimization and depth regularization. *Real-time Update* ensures the D.E. takes the latest Gaussian geometry as a conditioning signal, providing timely geometric constraints for Gaussian training.

## 5 Conclusion

In this paper, we presented $D^2GS$, a novel LiDAR-free framework for reconstructing dynamic urban street scenes. By effectively leveraging geometric priors derived from images, $D^2GS$ eliminates the dependency on LiDAR data, thus significantly simplifying data acquisition for autonomous driving applications. The core ideas of $D^2GS$ include: 1) An initialization strategy using multi-view depth estimation followed by a progressive training and pruning process, which efficiently generates a compact and representative initial Gaussian set. 2) A joint optimization approach employing a diffusion-based depth enhancer, which iteratively refines both the depth estimates and the Gaussian representation, providing robust geometric supervision throughout training. 3) The integration of a dedicated road node into the scene graph GS representation, explicitly modeling the ground plane using strong geometric priors. Extensive experimental results on the Waymo dataset validate the efficacy of our LiDAR-free reconstruction pipeline. While $D^2GS$ successfully eliminates the need for LiDAR, it still relies on accurate camera poses. In future work, we will investigate pose-free implementations to further simplify the pipeline and broaden its applicability.

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

# Supplemental Material

## A  Implementation Details

**Initialization:**  Our pipeline begins by generating initial depth estimates using DepthSplat [21] (weight name: "depthsplat-gs-base-dl3dv-256x448-randview2-6-d94d996f.pth") , which processes input images and camera parameters to produce per-view depth maps. These depth maps are then unprojected into 3D space using camera intrinsics and extrinsics to form a dense point cloud. We initialize static Gaussians from this point cloud, and then perform low-resolution training at a resolution of $240 \times 160$ for three 3,000-iteration pruning epochs. Through our progressive pruning strategy, we gradually filter out less significant Gaussians based on opacity values (the threshold is computed from percentiles of all Gaussians), ultimately retaining a subset of $2 \times 10^6$ Gaussians. To mitigate the impact of initial depth inaccuracies, we defer the activation of the Depth Enhancer module until the 1,000th iteration after the first epoch. The final rendered depth maps from this initialization phase (denoted as $\mathbf{D}^{\text{prev}}$) serve as the foundation for subsequent full training processes. For Road Node initialization, we use the road mask to identify the corresponding Gaussians and assign them to the Road Node.

**Training Details:**  The model contains 60,000 iterations of training, divided into 40,000 iterations for RGB-only training and 20,000 iterations for iterative Gaussian and depth enhancement training. The reason why we do not perform depth enhancement in the first 40000 iteration is that, the early training stages typically produce meaningless depth and image outputs. There are severe floaters and artifacts resulting from the aggressive densification process. We follow the default learning rate settings for Gaussian properties as established in OmniRe [1]. Our proposed Road Node's learning rate is 0.05. The Depth Enhancer module is activated halfway through training, with the diffusion process consisting of 80 steps and an early stop mechanism that prevents the smoothness loss from increasing for 8 continuous steps. For the warping loss, we utilize the views from 6 closest timestamp of the same camera ID. For quantitative comparisons, results for 3DGS, EmerNeRF, and S3GS are adopted from the S3GS [7] paper. All other methods were evaluated by us using their official codebases on a single NVIDIA H20 GPU. Key hyperparameters for our method are detailed in Tab. 4.

Table 4: Value of the key hyperparameters in our method.

| Hyperparameter | Value | Description |
|:---:|:---:|:---|
| $p \times p$ | $\frac{H}{8} \times \frac{H}{8}$ | Patch size for depth downsampling |
| $\lambda_{\text{C}}$ | 2.0 | Confidence threshold in Depth Enhancer |
| $\lambda_{\text{ref}}$ | 1.0 | Weight for reference depth loss in Depth Enhancer |
| $\lambda_{\text{smooth}}$ | $\frac{1}{8}$ | Weight for depth smoothness loss in Depth Enhancer |
| $\lambda_w$ | $\frac{1}{16}$ | Weight for warping loss in Depth Enhancer |
| $N$ | 20 | Depth Enhancer update frequency |
| $\lambda_{\text{normal}}$ | 10.0 | Weight for road normal alignment loss in Road Node |
| $\lambda_{\text{flat}}$ | 1.0 | Weight for road flatness loss in Road Node |
| $\lambda_{\text{road}}$ | 0.1 | Weight for Road Node loss |

## B   Datasets and Baselines:

Our main experiments are conducted on the Waymo NOTR Dynamic32 [27] dataset, comparing our method against S3GS [7], PVG [8], OmniRe [1], and LiDAR-free baselines. The specific segment IDs are listed in Tab. 5. Our ablation studies are performed on a 12-sequence subset of the Waymo NOTR dataset, with segment IDs provided in Tab. 6.

Table 5: Segment IDs of Waymo NOTR Dynamic32 Dataset.

| | | | | | | | |
|---|---|---|---|---|---|---|---|
| seg102319... | seg103913... | seg104444... | seg104980... | seg105887... | seg106250... | seg106648... | seg109636... |
| seg110170... | seg117188... | seg118463... | seg119178... | seg119284... | seg120278... | seg121618... | seg122514... |
| seg123392... | seg148106... | seg168016... | seg181118... | seg191876... | seg225932... | seg254789... | seg441423... |
| seg508351... | seg522233... | seg583504... | seg624282... | seg767010... | seg882250... | seg990779... | seg990914... |

Table 6: Segment IDs of a 12 sequences subset used in our ablation experiments.

| | | | | | |
|---|---|---|---|---|---|
| seg102319... | seg104980... | seg105887... | seg109636... | seg110170... | seg117188... |
| seg119284... | seg122514... | seg123392... | seg168016... | seg191876... | seg441423... |

# C Additional Results

In this section, we present more qualitative comparisons of our methods with others. Same as the main submission, we show the results of the LiDAR supervised methods (S3GS [7], PVG [8], OmniRe [1], and LiDAR-free methods (OmniRe w/o LiDAR, OmniRe + DepthSplat, ours).

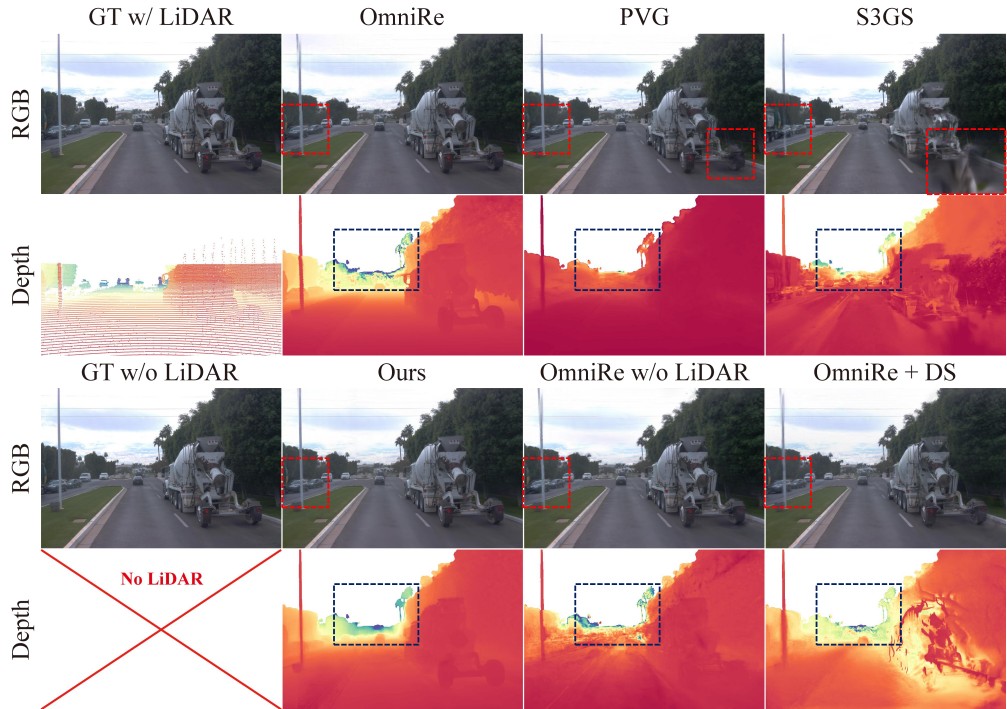

Figure 5: Comparison of our method with S3GS, PVG, OmniRe, and LiDAR-free baselines.

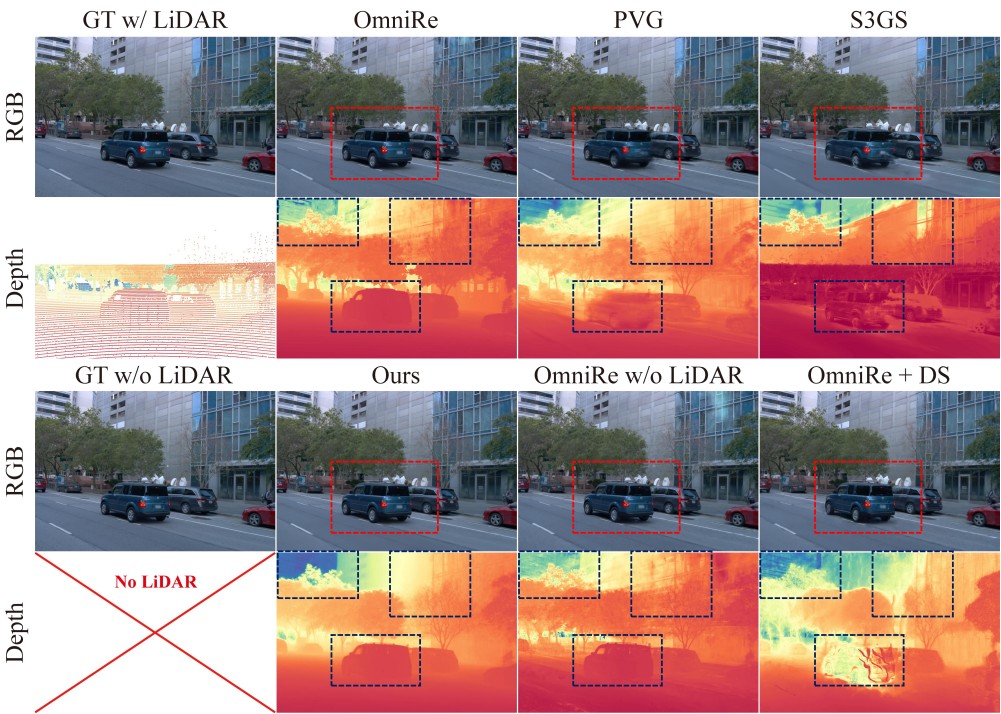

Figure 6: Comparison of our method with S3GS, PVG, OmniRe, and LiDAR-free baselines.

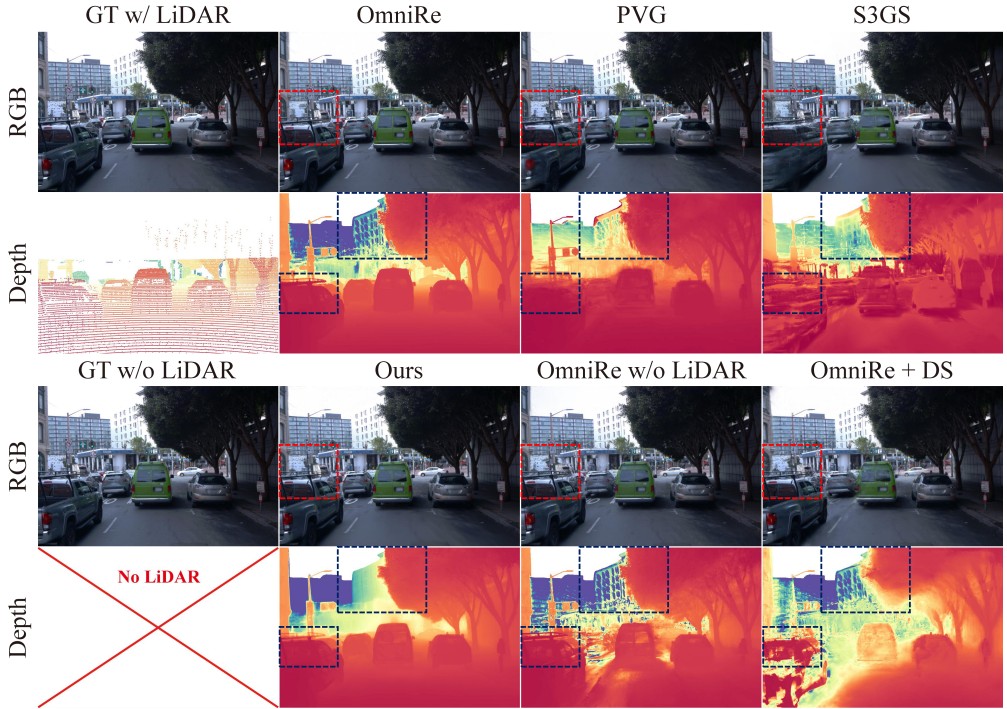

Figure 7: Comparison of our method with S3GS, PVG, OmniRe, and LiDAR-free baselines.

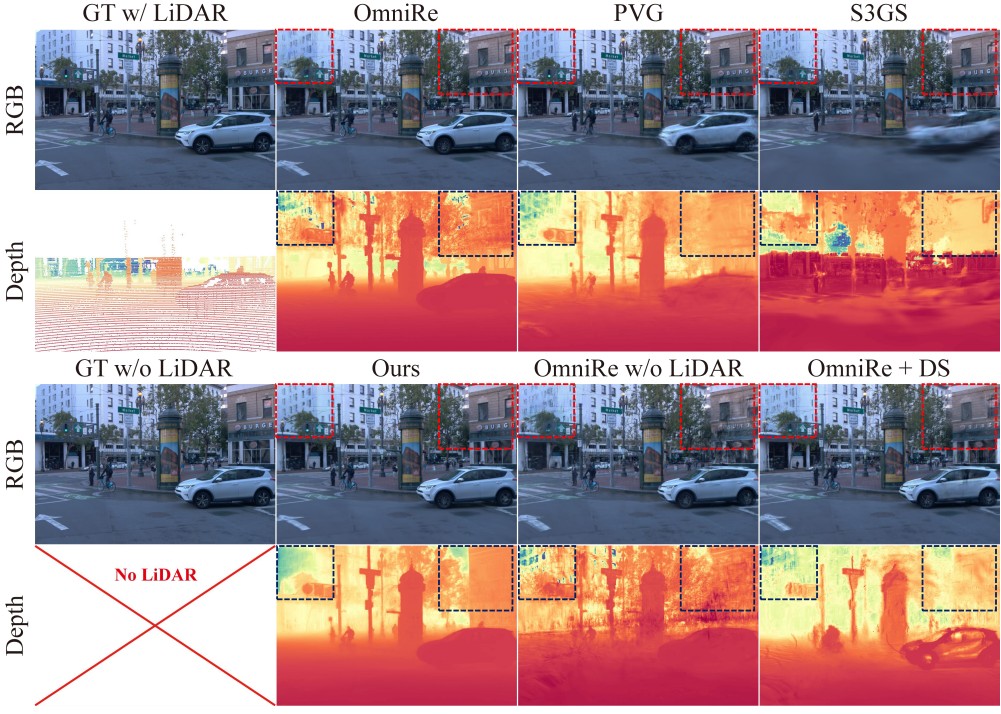

Figure 8: Comparison of our method with S3GS, PVG, OmniRe, and LiDAR-free baselines.

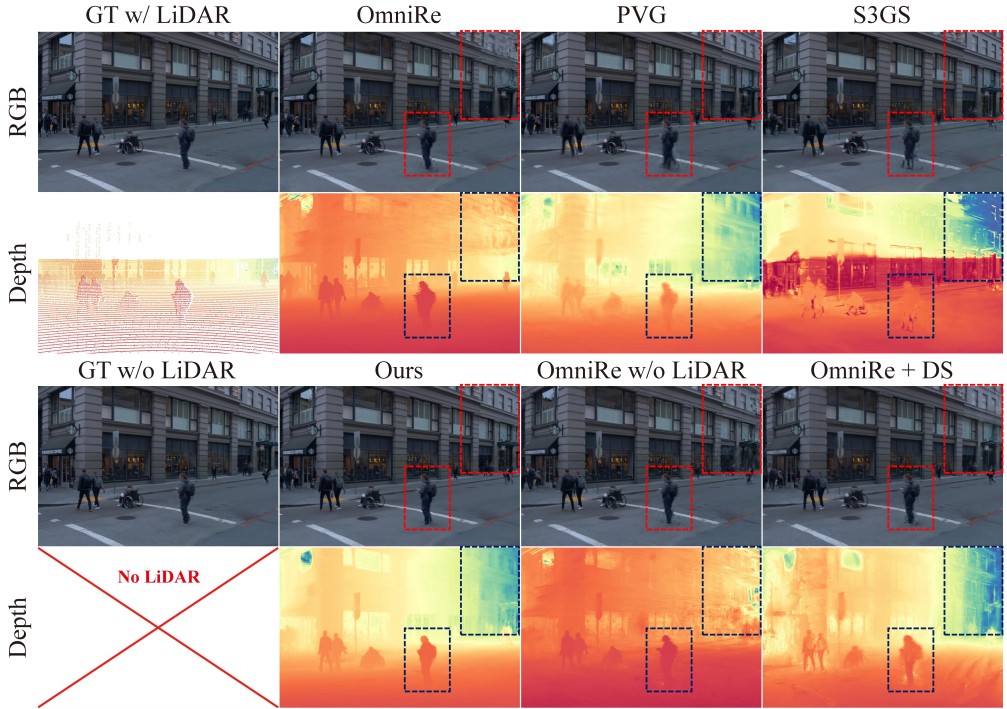

Figure 9: Comparison of our method with S3GS, PVG, OmniRe, and LiDAR-free baselines.

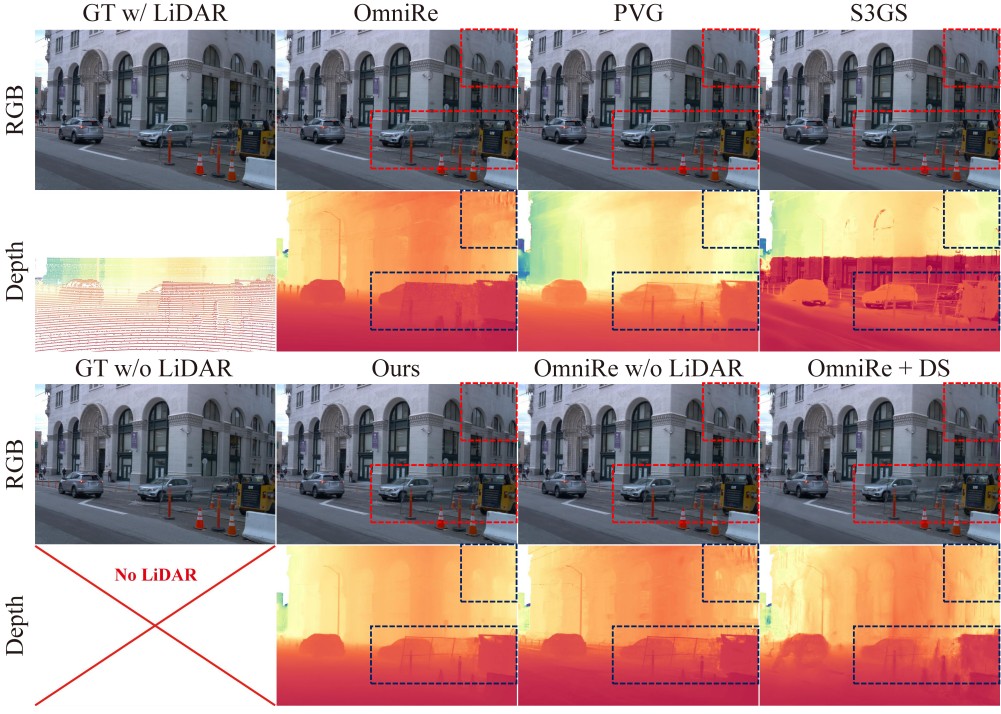

Figure 10: Comparison of our method with S3GS, PVG, OmniRe, and LiDAR-free baselines.

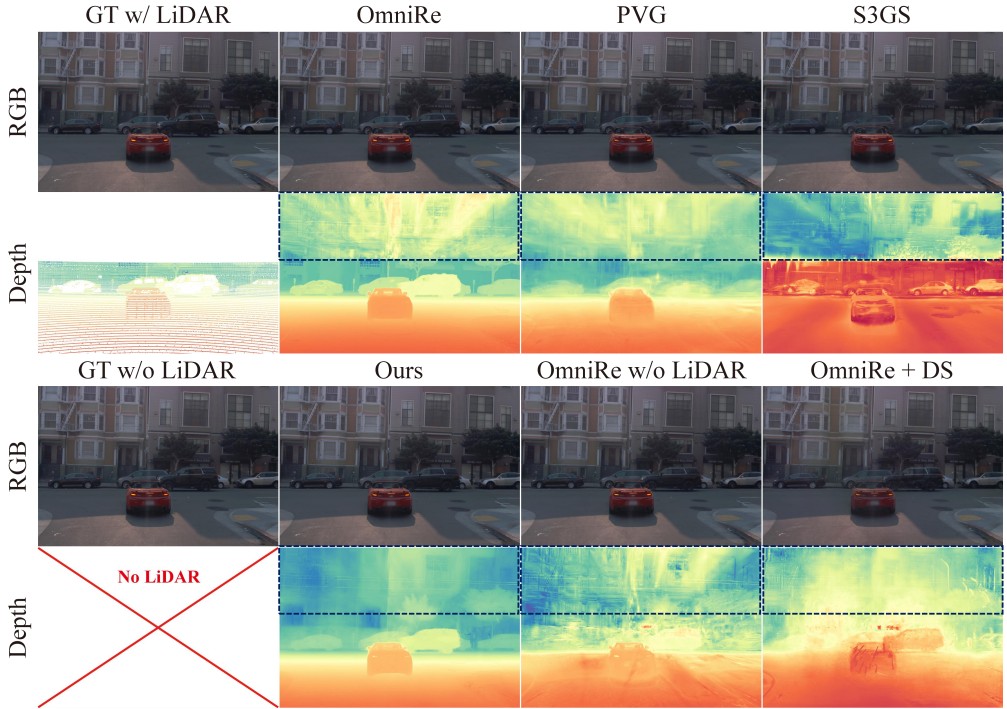

Figure 11: Comparison of our method with S3GS, PVG, OmniRe, and LiDAR-free baselines.

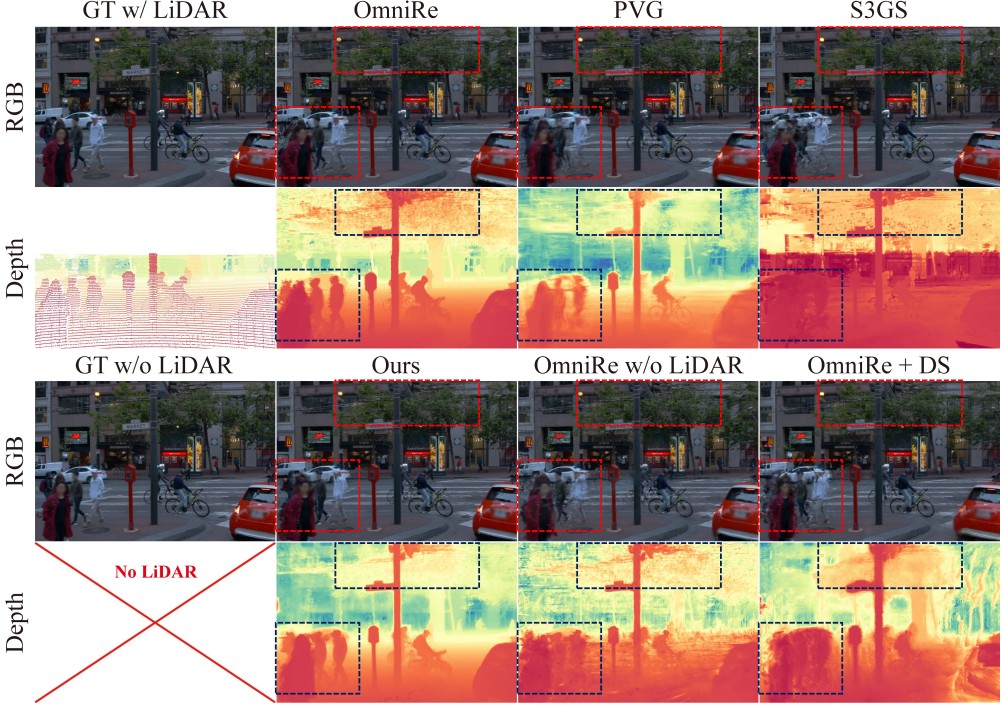

Figure 12: Comparison of our method with S3GS, PVG, OmniRe, and LiDAR-free baselines.

# D  Additional Experiments and Analyses

**Summary.** This section expands the computational analysis, clarifies assumptions about camera poses and our handling of dynamic objects, details sky modeling, strengthens ablations (including image-level metrics and road-node losses), visualizes depth evolution during training, and analyzes the initialization choices (SfM vs. MVS vs. monocular metric depth).

**Notation.** We use the main-paper notation throughout. "P.P." denotes *Progressive Pruning*, "D.E." denotes the *Depth Enhancer*, and "R.N." denotes the *Road Node*. NVS denotes novel-view synthesis.

## D.1  Computational Analysis of Progressive Pruning

Dense depth initialization yields an extremely large set of back-projected points. Training Gaussians on these points is both redundant and error-prone, because abundant low-accuracy points often produce artifacts. P.P. therefore serves two goals: (i) reduce computation and memory by early removal of inconsistent Gaussians and (ii) deliver a higher-quality initial point cloud that improves reconstruction.

**Computational Analysis.** Tab. 7 profiles memory, runtime, and the evolution from initial to retained training points over the first 30k iterations. The results show that when the initial input point cloud contains as many as 15M points, the Gaussian model retains around 6M points that are not effectively pruned. In contrast, our P.P. method prevents a significant waste of computational resources and avoids efficiency bottlenecks. Even with very large initial sets, P.P. can avoid out-of-memory errors and shorten training by eliminating redundant, inconsistent Gaussians before full optimization. In contrast, the original pruning method prunes primarily during rendering optimization, which defers these savings.

Table 7: Analysis of computational cost, comparing our P.P. method with a baseline using a manageable number of points to demonstrate the efficiency gains.

| Method | Initial Points | Training Points$^{\dagger}$ | GPU RAM | Runtime |
|---|---|---|---|---|
| w/o P.P. | 15M | 6M - 8M | 34 GB | 10 h |
| P.P. | 2M | 2M - 3M | 20 GB | 8 h |

$^{\dagger}$Retained points used in optimization within the first 30k iterations.

## D.2  The Dependency on Camera Pose Accuracy

Despite the LiDAR calibration issue which we aim to solve in our method, we assume accurate camera poses, consistent with common urban driving scene reconstruction settings and dataset priors.

To further address inaccurate camera poses, a common issue in data calibration, we propose the following possible solutions for future work. Specifically, when precise camera poses are unavailable, pose estimation algorithms, such as learning-based methods or traditional SfM, can provide initial estimates. Subsequently, a joint optimization module for camera poses can be incorporated during reconstruction, enabling simultaneous refinement of both the poses and the reconstructed scene.

## D.3  Modeling Dynamic Objects

Our method can handle dynamic objects properly through the following aspects:

**Scene graph decomposition.** We model and decompose dynamic objects using a Scene Graph, an approach that follows our baseline, OmniRe [1]. Each dynamic object is represented as an independent node within this graph, equipped with its own set of Gaussian models (for dynamic or rigid representations).

**Depth refinement for dynamics.** Initial MVS depths inevitably contain errors on moving objects. First, since our method uses given ground-truth camera poses from Waymo, dynamic objects do not affect pose estimation. Depth errors are therefore localized to these dynamic areas, with minimal

impact on the static background. Furthermore, during the iterative optimization of the Gaussians and the D.E., the depth of dynamic objects is further refined (as shown in Fig. 13). Specifically, this refinement is driven by both the photometric loss from Gaussian splatting and the accurate geometric supervision provided by our diffusion-based D.E.. This joint optimization loop is effective for both static and dynamic scene components, enabling correction of initial errors and more accurate depth prediction.

## D.4    Detailed Ablations with Image-Level Metrics

We complement geometric ablations with image reconstruction and NVS metrics, with results shown in Tab. 8. The full model performs best across settings, corroborating Tab. 3 in the main paper: degraded depth quality typically yields poorer NVS. Notably, the Periodic Update variant attains high reconstruction quality but markedly worse NVS, suggesting overfitting to training-view photometry with stale depth. This also underscores the importance of our real-time iterative refinement.

Table 8: Image-level metrics for ablations (image reconstruction and NVS). "Update" denotes D.E. update scheme.

| Components | | | Settings of D.E. | | | | Image reconstruction | | | Novel view synthesis | | |
|---|---|---|---|---|---|---|---|---|---|---|---|---|
| R.N. | P.P. | D.E. | $L_{\text{ref}}$ | $L_w$ | $L_{\text{smooth}}$ | Update | PSNR ↑ | SSIM ↑ | LPIPS ↓ | PSNR ↑ | SSIM ↑ | LPIPS ↓ |
| ✗ | ✓ | ✓ | ✓ | ✓ | ✓ | **Real-time** | 32.67 | 0.939 | 0.087 | 27.67 | 0.854 | 0.139 |
| ✓ | ✗ | ✓ | ✓ | ✓ | ✓ | **Real-time** | 32.25 | 0.932 | 0.099 | 27.17 | 0.847 | 0.149 |
| ✓ | ✓ | ✗ | ✗ | ✗ | ✗ | **Real-time** | 32.77 | 0.939 | 0.087 | 27.52 | 0.851 | 0.143 |
| ✓ | ✓ | ✓ | ✓ | ✗ | ✗ | **Real-time** | 32.57 | 0.940 | 0.086 | 27.29 | 0.852 | 0.140 |
| ✓ | ✓ | ✓ | ✓ | ✓ | ✓ | **Periodic** | 33.95 | 0.951 | 0.075 | 26.90 | 0.823 | 0.128 |
| ✓ | ✓ | ✓ | ✓ | ✓ | ✓ | **Real-time** | 33.63 | 0.947 | 0.079 | 28.04 | 0.863 | 0.130 |

## D.5    Road Node: Loss Analysis

We ablate the two R.N. loss terms independently, and conduct experiments on a subset of Waymo dataset (sequence id: '031','035', '084', '089', '102', '111', '222', '382'). As shown in Tab. 9, both the loss $L_{\text{plane}}$ and the loss $L_{\text{shape}}$ contribute positively to a more accurate geometry (evaluated with depth results). The best performance is achieved when both are used together.

Table 9: Ablation of road-node losses on Waymo subset.

| Method | L1 ↓ | Abs. Rel. ↓ | RMSE ↓ | $\delta < 1.25$ ↑ |
|---|---|---|---|---|
| w/o $L_{\text{plane}}$ | 2.7879 | 0.1395 | 5.0075 | 0.9270 |
| w/o $L_{\text{shape}}$ | 2.6502 | 0.1211 | 5.0192 | 0.9320 |
| **Full loss** | 2.4245 | 0.1161 | 4.5535 | 0.9459 |

## D.6    Depth Evolution Across Training

Fig. 13 visualizes the online refinement process by showing: (i) the Gaussian-rendered depth (D.E. input), (ii) the refined depth (D.E. output), and (iii) visualizations of this process at three checkpoints (initial, mid, and final training stages). We observe a clear removal of floating fragments, stabilization of large planar regions, and improved cross-view geometric consistency as the refinement progresses.

## D.7    Initialization: SfM vs. MVS vs. Monocular

In this section, we give a more comprehensive analysis of different initialization methods. Existing depth estimation methods are inevitably imperfect and unsuitable to use as strict ground-truth for high-fidelity reconstruction. As discussed in the main paper, monocular methods often yield relative depth with unknown scale, while MVS methods can struggle in dynamic scenes. Our core contribution is a framework that *starts* from an imperfect *metric* depth prior and jointly optimizes geometry and depth. Metric initialization is crucial for scale-correct point-cloud seeding and for strong, multi-view-consistent supervision.

|  Initial  |  Final  |
| :---: | :---: |

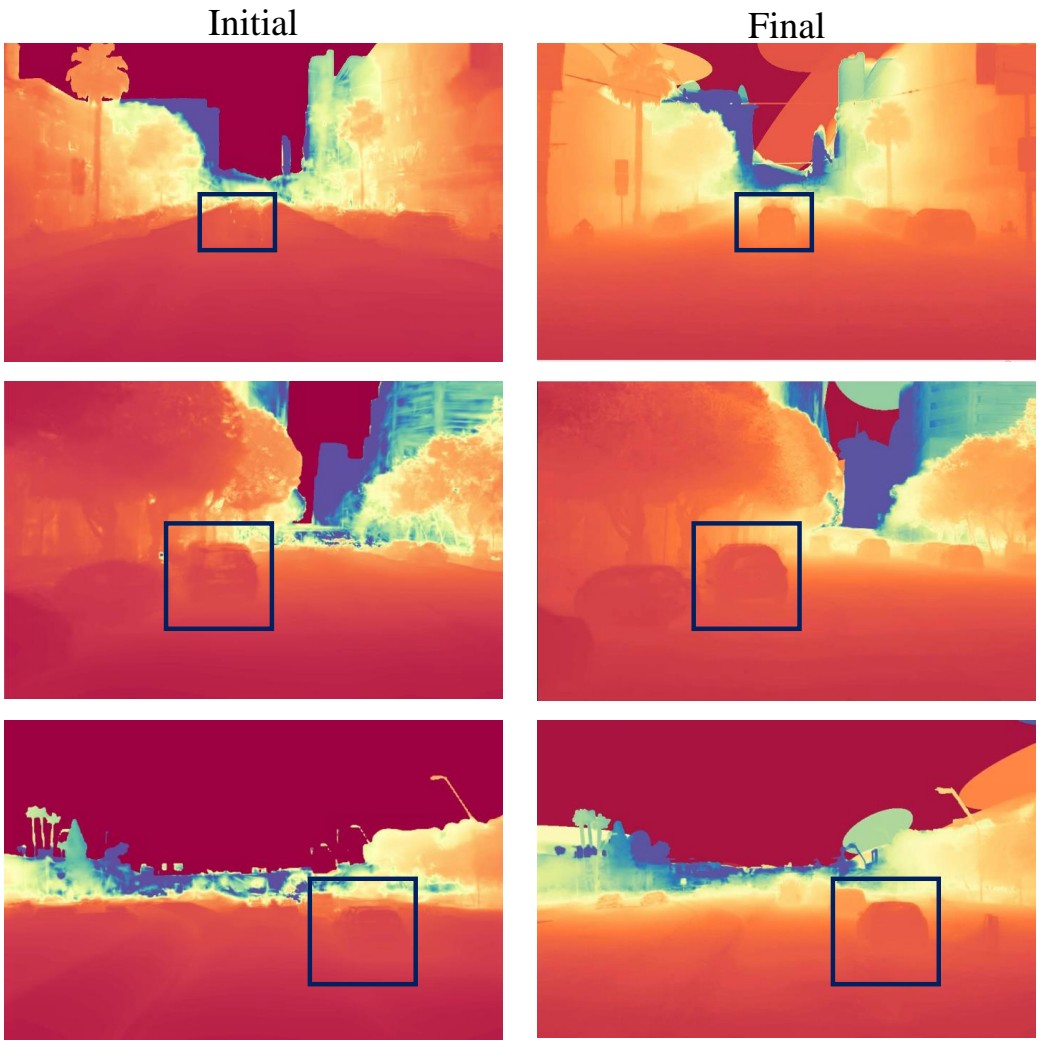

Figure 13: Depth evolution with online updates in three different stages: initial → final.

Recent monocular depth estimation foundation models (e.g., UniDepthV2 [38] and Metric3D [39]) also report strong metric depth prediction performance. We adopt a DepthSplat-style MVS depth estimator in our main paper. Nevertheless, our pipeline is orthogonal to the depth source: any reasonable *metric* initializer can be used.

**Monocular metric depth (UniDepthV2) in place of MVS.** We replace the DepthSplat MVS method with UniDepthV2 [38] to initialize our pipeline. We conduct experiments on a subset of Waymo dataset (sequence id: '031','035', '084', '089', '102', '111', '222', '382'), and Tab. 10 shows that our method still improves over baselines, confirming orthogonality to the initializer. Gains are smaller than with DS, likely because UniDepthV2 provides stronger initial depths and thus less headroom for refinement.

**Why not SfM: a sparse point cloud initialization.** We define dense initialization as back-projecting *pixelwise* metric depth maps into a redundant but coverage-complete point cloud. Sparse initialization denotes sampled point clouds (e.g., LiDAR, COLMAP), which provide only sparse supervision and may miss fine-scale or specific parts of the geometry.

In urban driving scenes, SfM still tends to be sparse due to limited overlap, textureless regions, and occlusions. Our contributions align naturally with dense initialization: P.P. compresses dense

Table 10: Monocular metric depth estimation as an initialization alternative on Waymo subset.

| Methods | Image reconstruction | | | Novel view synthesis | | |
|---|---|---|---|---|---|---|
| | PSNR | SSIM | LPIPS | PSNR | SSIM | LPIPS |
| OmniRe | 33.96 | 0.955 | 0.057 | 28.50 | 0.874 | 0.117 |
| OmniRe+DS | 31.30 | 0.936 | 0.078 | 28.11 | 0.876 | 0.110 |
| OmniRe+UniDepthV2 | 32.73 | 0.944 | 0.068 | 29.67 | 0.885 | 0.098 |
| Ours+DS | 34.35 | 0.954 | 0.068 | 29.75 | 0.883 | 0.107 |
| Ours+UnidepthV2 | 34.39 | 0.954 | 0.069 | 29.92 | 0.891 | 0.096 |

initialization priors before full optimization, and D.E. requires dense targets for per-pixel supervision. However, with minor changes (e.g., drop P.P. and seed D.E. with reprojected sparse depths), the pipeline still applies to SfM sparse initialization.

