# OpenReview forum: "D$^2$GS: Dense Depth Regularization for LiDAR-free Urban Scene Reconstruction"
_NeurIPS.cc/2025/Conference — NeurIPS 2025 poster_

### Official Review · Reviewer_GUm1 · 2025-06-24

**Clarity:** 2
**Significance:** 3
**Originality:** 2
**Rating:** 4
**Confidence:** 5

**Summary:**

This paper present the a LiDAR-free framework for reconstructing dynamic urban street scenes named D^2GS. The key components of this paper involve: 1) using multi-view depth estimation to initialize the 3DGS; 2) employ the Depth-Enhancer to refines both the depth estimates and Gaussian geometric information ;3) apply several regularization items to model road region. The results on Waymo seem to validate the efficacy of the LiDAR-free reconstruction pipeline.

**Questions:**

*  Moreover, the adoption of multi-view depth estimation inherently struggles to handle dynamic objects, so what is implantation details of this method to estimate the dynamic image sequence.

* As the sky region has the infinite distance which is difficult to initialize with the point cloud. Although I notice the  semantic mask in Figure.2 (main paper), how does the proposed method  model the sky?

**Ethical Concerns:**

["NO or VERY MINOR ethics concerns only"]

**Final Justification:**

I think the rebuttal have addressed my concerns and i would like to improve my rating

**Limitations:**

yes

**Quality:**

3

**Strengths And Weaknesses:**

## Strengths :
1) D2GS method consistently outperforms all other methods by a large margin in Table 1  and Figure 3 both in Image reconstruction and NVS quality.

2) The lidar-free method designed for autonomous driving scenario have more practical application

3) The diffusion-based Depth Enhancer Module is interesting and somehow innovative, which simultaneous improve the surprised signal (estimated depth) and 3D Gaussian presentation.

4) The Road Node decomposition make sense and help improve reconstruction quality in qualitative results in Figure 4.

## Weaknesses:
* Lines 51–55: The motivation of this paper is not clearly presented, particularly regarding why monocular depth estimation is considered unsuitable for dynamic scenes. Existing approaches, such as UniDepth and Metric3Dv2, have already demonstrated strong performance in estimating depth in street-view scenarios. Furthermore, the reason for adopting a DepthSplat-based multi-view stereo (MVS) approach, instead of leveraging monocular depth estimation, remains insufficiently justified.

* It is unclear why a heuristic threshold is used during initialization to reduce the number of Gaussians, given that 3D Gaussian Splatting (3DGS) inherently supports pruning based on radius, opacity, and scale. what is the problem if the proposed method apply the  ADC in vanilla 3DGS?

* As a study focused on street scene reconstruction, an important component is how to handle the dynamic agents. However, I could not find relevant details regarding this issue either in the supplementary materials or in the main paper.

* The paper merely attributes performance degradation to the use of DepthSplat-generated depth maps, without exploring alternative depth estimation methods such as monocular depth which seems to be insufficient to support the proposed method.



* typos: Line 134  ‘..’   Line135  uing->using  Lin57 TO -> to

---

> ### Author Rebuttal · Authors · 2025-07-31
>
> ### **Dear Reviewer GUm1,**
>
> We sincerely thank Reviewer GUm1 for the insightful review and valuable feedback. In this rebuttal, we provide new experiments using a monocular metric depth estimator to demonstrate our framework's robustness, alongside detailed clarifications on the motivation for our key design choices.
>
> ### **1\. W1 & W4: What is the motivation of the current depth initialization method? And what about alternative monocular metric depth estimation methods?**
>
> We thank you for the important questions regarding our motivation for choosing depthsplat-based MVS as our initialization method.
>
> **From our perspective, existing depth estimation methods still produce imperfect results, making them unsuitable for direct use as ground truth in high-quality reconstruction tasks.** As we argued in L51-55, monocular depth estimation methods typically generate relative depth with an unknown scale, while stereo-based methods struggle with dynamic scenes. Therefore, we aim to design a framework to take the imperfect, metric depth maps as a starting point and jointly optimize both the scene geometry and the depth, which is also our core contribution. We choose **metric depth** specifically because it is essential for **initializing point clouds** at the correct scale and for enabling **strong, direct, and multi-view consistent geometric supervision** during training.
>
> Second, with the rapid development of depth foundation models, recent works such as UniDepth and Metric3D as you suggested, have achieved impressive performance in metric depth estimation. We adopt a DepthSplat-like MVS approach mainly because the camera poses are known in our setting. However, we emphasize that our framework is **orthogonal** to the choice of initial depth source and can be generalized to any method that provides a reasonable metric depth initialization, whether it originates from MVS or monocular metric depth estimation.
>
> Here, we provide experiments using the state-of-the-art monocular metric depth estimator,  **UniDepthV2 \[1\]**, to initialize our pipeline instead of MVS. The results on the Waymo dataset are as follows:
>
> **Tab. A**: Supplementary comparison to Tab. 1 with quantitative results of different initialization and training methods. These results are based on 8 sequences from the Waymo dataset due to the time limitations of the rebuttal phase. We will provide the full results in the revised manuscript.
> | **Methods** | **Image reconstruction** | | | | **Novel view synthesis** | | |
> | :--- | :---: | :---: | :---: | :---: | :---: | :---: | :---: |
> | | **PSNR**$\uparrow$ | **SSIM**$\uparrow$ | **LPIPS**$\downarrow$ | | **PSNR**$\uparrow$ | **SSIM**$\uparrow$ | **LPIPS**$\downarrow$ |
> | OmniRe | 33.39 | 0.952 | 0.063 | | 27.89 | 0.864 | 0.133 |
> | OmniRe+DS | 31.99 | 0.943 | 0.070 | | 28.26 | 0.871 | 0.115 |
> | OmniRe+UniDepth | 32.87 | 0.949 | 0.062 | | 29.13 | 0.880 | 0.103 |
> | Ours+DS | 33.93 | 0.947 | 0.078 | | 29.25 | 0.879 | 0.121 |
> | Ours+Unidepth | 33.58 | 0.953 | 0.072 | | 29.42 | 0.888 | 0.111 |
> ||
>
> As shown in Tab. A, the results demonstrate that our framework can effectively leverage and refine the initial depth from a monocular method. This confirms that our proposed optimization pipeline can be generalized to other specific choice of initializer. However, the performance improvement of ours + UniDepthV2 is less significant than that of ours + DS, compared to the OmniRe counterpart. We infer that this is because UniDepthV2 provides more accurate depth estimation, leaving less room for further improvement by our method. We will add these experiments and this discussion to our revised manuscript.
>
> \[1\] *Piccinelli, Luigi, et al. "Unidepthv2: Universal monocular metric depth estimation made simpler." arXiv preprint arXiv:2502.20110 (2025).*
>
> ### **2\. W2: What is the difference between our Progressive Pruning method and ADC in vanilla 3DGS?**
>
> Thank you for this question, which allows us to clarify the critical role of our Progressive Pruning component and its fundamental differences from the native pruning within 3DGS.
>
> * **Distinct Goals:**
>   1. The goal of ADC in vanilla 3DGS is to refine rendering quality **during** the optimization by splitting Gaussians in high-gradient areas and removing those with near-zero opacity. In contrast, our Progressive Pruning component is a pre-processing step designed to **(1) ensure computational feasibility** by efficient point pruning and **(2) establish a geometrically accurate and compact starting point** for the main training.
> * **Why Progressive Pruning is Necessary:**
> 1. **Computational Feasibility:** Initializing from dense depth maps generates an enormous number of points. Attempting to load this point cloud into GPU memory for standard training pipeline **may cause a GPU OOM** error at the very first iteration. The experiment results in Table B show that loading 15 millions of points leads to the model maintaining around 6 million points during training that cannot be effectively pruned, indicating significant **point redundancy** and **a slowdown in training speed**.
> 2. **Geometric Distillation:** More importantly, Progressive Pruning is a scheduled, multi-stage pre-training process that acts as a Geometric Distillation mechanism. First, this process operates on the massive point cloud by **completely disabling densification operations** like splitting and cloning. The learning objective is focused exclusively on refining the geometry. Within this simplified framework, a Gaussian's opacity is optimized to reflect its consistency with the geometric constraints.  Second, the progressive filtering is guided by a final target point count, not a fixed threshold. As mentioned by L344 in our Supplementary Material, in each stage, we rank all Gaussians by their learned opacity and prune those that contribute least to a coherent geometric structure. This **adaptive, ranking-based selection** ensures that a compact and representative set of Gaussians is preserved **at a specific amount**.
>
>
> **Table B:** Analysis of computational cost, comparing our Progressive Pruning method with a baseline using a manageable number of points to demonstrate the efficiency gains.
> | Method   | Initial Point num.   | Training Point num.* (first 30k iter) | GPU RAM | Runtimes |
> |:--------:|:--------------------:|:-----------------------------------:|:-------:|:--------:|
> | w/o P.P. | $1.5×10^7$                | $6×10^6$ - $8×10^6$                             | 34G     | 10h      |
> | P.P      | $2×10^6$                | $2×10^6$ - $3×10^6$                           | 20G     | 8h       |
> | |
>
>
> ### **3\. W3 & Q1: How does this paper handle dynamic objects?**
>
> Thank you for pointing out the lack of clarity regarding our handling of dynamic objects. We will add a dedicated section in the final manuscript to elaborate on these technical details. In response to your specific questions:
>
> * **Dynamic Object Modeling:** Our approach to modeling and decomposing dynamic objects follows our baseline, OmniRe, by utilizing a Scene Graph. Each dynamic object is represented as an independent node within this graph, possessing its own set of Gaussian models (e.g., dynamic or rigid representations).
> * **Depth Estimation for Dynamic Objects:** We acknowledge that initial depth estimates derived from MVS will inevitably contain errors for moving objects, a common challenge in depth estimation. First, since our method uses given camera poses from Waymo, dynamic objects do not affect pose estimation. Depth errors are therefore localized to these dynamic areas, with minimal impact on the static background. Furthermore, during the iterative optimization of the Gaussians and the Depth Enhancer, the depth of dynamic objects is further refined. Specifically, this refinement is driven by both the photometric loss from Gaussian splatting and the accurate geometric supervision provided by our diffusion-based Depth Enhancer. This joint optimization loop is **equally effective for both static and dynamic scene components**, allowing it to effectively correct initial errors and achieve accurate depth prediction
>
> We believe a comprehensive visualization of the depth refinement process would best illustrate the optimization of depth for dynamic objects. As requested by Reviewer e8Wz, we will include such visualizations and a supplementary explanation in the final manuscript to demonstrate the rationale and effectiveness of our method.
>
> ### **4\. Q2: How does this paper handle the sky?**
>
> Thank you for this observation. We follow the standard practice used in OmniRe to model the sky. Specifically, we use a pre-processed sky-mask for calculating **opacity loss**, encouraging Gaussians in the sky region to become fully transparent. The appearance of the sky itself is **handled by a separate model** (e.g., a cubemap representation), which is decoupled from the 3D scene Gaussians. We will clarify this in the implementation details of our final manuscript.

---

### Official Review · Reviewer_e8Wz · 2025-07-02

**Clarity:** 3
**Significance:** 3
**Originality:** 3
**Rating:** 5
**Confidence:** 3

**Summary:**

This paper proposes a LiDAR-free Gaussian Splatting reconstruction method for urban scenes, named D^2GS.
First, the authors employ a multi-view depth estimation network to generate dense initial point clouds and reduce their quantity through a progressive pruning strategy.
Second, using rendered depth as reference, they online-update a diffusion-based depth estimation model to produce scale-aligned metric depth for geometric supervision.
Finally, to enhance road surface reconstruction capability, a road node regularization scheme is introduced.
Both quantitative and qualitative results demonstrate that the proposed method achieves state-of-the-art performance using only images as input.

**Questions:**

1. Although LiDAR point clouds are not available, it is still possible to generate an initial point cloud using SfM. In fact, most image-only GS methods use SfM point clouds for initialization. Could you explain why dense depth estimation is used for initialization instead of SfM point clouds?
2. The optimization of road node includes two different loss functions, but there is no independent ablation for both. Can the impact of both on performance be further analysed quantitatively and qualitatively?
3. The ablation study only reports depth-related metrics. However, for GS scene reconstruction, rendering performance is also important. Could you report the impact of each component on novel view synthesis (NVS) performance?
4. One of the core contributions of this paper is the online-updated depth supervision. However, there is no visualization for this part. Could you provide visualizations of the estimated depth and rendered depth at different stages to demonstrate the effect of online updates?

**Ethical Concerns:**

["NO or VERY MINOR ethics concerns only"]

**Final Justification:**

The authors have adequately addressed my concerns regarding initialization and ablation studies. I believe the innovative method proposed in this paper contributes to improving the geometric representation of 3DGS in urban scenes, and I am therefore willing to raise my score.

**Limitations:**

yes

**Paper Formatting Concerns:**

Line 134, there is an extra period.

**Quality:**

3

**Strengths And Weaknesses:**

Strengths:

1. The proposed reconstruction method eliminates the need for LiDAR sensor data, reducing the data collection costs for urban scene reconstruction while avoiding the challenges associated with precise calibration in multi-sensor systems. This advancement contributes positively to promoting large-scale applications in the field.
2. The writing and equations are clear and easy to understand.

Weaknesses:

1. The motivation for using estimated depth from multi-view as initialization is not clearly explained.
2. The ablation experiments of some modules are not sufficient and their effects are not clear enough.

---

> ### Author Rebuttal · Authors · 2025-07-31
>
> ### **Dear Reviewer e8Wz,**
>
> We sincerely thank Reviewer e8Wz for the insightful review and valuable suggestions. In this rebuttal, we clarify our motivations and provide comprehensive ablation studies to quantitatively analyze each component's impact.
>
> ### **1\. W1 & Q1: Could you explain why dense depth estimation is used for initialization instead of SfM point clouds?**
> Thank you for this important question. We acknowledge that SfM is a common initialization strategy for many image-only GS methods. Our decision to use dense depth maps from a MVS approach instead of SfM was based on the following considerations for our specific problem setting:
>
> 1. **Availability of High-Quality Camera Poses:** In our primary dataset (Waymo), high-quality, pre-calibrated **camera poses are already provided**. A key function of SfM is to solve for these poses from images, which is not only redundant in our case but also computationally intensive, especially for large-scale urban scenes with lots of images.
> 2. **Requirement for Metric Depth:** Our method is designed to leverage and refine dense geometric priors with a correct metric scale. Traditional SfM pipelines typically produce point clouds that have an arbitrary scale, which would require an additional, non-trivial step to align to a metric scale. Using an MVS-based method with known camera poses provides a more direct and efficient path to the dense, metric depth maps that serve as a strong geometric starting point for our framework.
> 3. **Orthogonality to Our Method:** our framework is orthogonal to the choice of initial depth source and can be generalized to any method that provides a reasonable metric depth initialization.
>
> In future work, we plan to extend our method to a camera-pose-free setting. In that scenario, integrating a pose estimation module, such as a learning-based method or a traditional SfM approach, would be a necessary and valuable first step, followed by joint optimization of the poses during reconstruction. We will clarify this motivation in the revised manuscript.
>
> ### **2\. W2 & Q2: The independent ablation for two different loss functions of Road Node.**
> Thank you for this valuable suggestion. We agree that a more detailed ablation is needed to demonstrate the individual contributions of the two loss functions for the road node. We have performed this additional ablation study as follows.
>
> Tab. A: Ablation study of the two loss functions for the road node. These results are based on 8 sequences from the Waymo dataset due to the time limitations of the rebuttal phase. We will provide the full results in the revised manuscript.
> | **Methods** | **L1**$\downarrow$ | **Abs. Rel.**$\downarrow$ | **RMSE**$\downarrow$ | **$\delta <$ 1.25**$\uparrow$ |
> | :--- | :---: | :---: | :---: | :---: |
> | w/o $L_{plane}$ | 2.7879 | 0.1395 | 5.0075 | 0.9270 |
> | w/o $L_{shape}$ | 2.6502 | 0.1211 | 5.0192 | 0.9320 |
> | Final version | 2.4245 | 0.1161 | 4.5535 | 0.9459 |
> ||
>
> As shown in Tab. A, both the loss $L_{plane}$ and the loss $L_{shape}$ contribute positively to a more accurate geometry (evaluated with depth results). The best performance is achieved when both are used together. We will include these quantitative results and a corresponding qualitative analysis in the final version of the manuscript.
>
> ### **3\. W2 & Q3: The ablation study only reports depth-related metrics. Could you report the impact of each component on novel view synthesis (NVS) performance?**
> Thank you for this excellent suggestion. We completely agree that novel view synthesis (NVS) metrics are crucial for evaluating scene reconstruction quality. Our original ablation study emphasized geometric metrics because the ill-posed nature of sparse-view reconstruction can be misleading, and models can easily overfit to training views, yielding high PSNR scores that don't reflect true geometric accuracy. We now have updated our main ablation study to include PSNR, SSIM, and LPIPS, and will add the following table to the manuscript.
>
> Tab. B: Image-level performance of all ablation study results in our original manuscript.
> | **R.N.** | **P.P.** | **D.E.** | **$L_{\text{ref}}$** | **$L_w$** | **$L_{\text{smooth}}$** | **Update** | **Image reconstruction** | | | **Novel view synthesis** | | |
> | :---: | :---: | :---: | :---: | :---: | :---: | :---: | :---: | :---: | :---: | :---: | :---: | :---: |
> | | | | | | | | **PSNR**$\uparrow$ | **SSIM**$\uparrow$ | **LPIPS**$\downarrow$ | **PSNR**$\uparrow$ | **SSIM**$\uparrow$ | **LPIPS**$\downarrow$ |
> | × | ✓ | ✓ | ✓ | ✓ | ✓ | Real-time | 32.67 | 0.939 | 0.087 | 27.67 | 0.854 | 0.139 |
> | ✓ | × | ✓ | ✓ | ✓ | ✓ | Real-time | 32.25 | 0.932 | 0.099 | 27.17 | 0.847 | 0.149 |
> | ✓ | ✓ | × | × | × | × | Real-time | 32.77 | 0.939 | 0.087 | 27.52 | 0.851 | 0.143 |
> | ✓ | ✓ | ✓ | ✓ | × | × | Real-time | 32.57 | 0.940 | 0.086 | 27.29 | 0.852 | 0.140 |
> | ✓ | ✓ | ✓ | ✓ | ✓ | ✓ | Periodic | 33.95 | 0.951 | 0.075 | 26.90 | 0.823 | 0.128 |
> | ✓ | ✓ | ✓ | ✓ | ✓ | ✓ | Real-time | 33.63 | 0.947 | 0.079 | 28.04 | 0.863 | 0.130 |
> |||
>
> As shown in the Tab. B, each of our proposed components contributes to the final performance. When combined with Table 3 in the main manuscript, the results show that, in most cases, poorer depth quality leads to inferior NVS performance.  Notably, the 'Periodic Update' variant shows a significant drop in NVS performance compared to our full model, indicating it has overfitted to the training views' photometry with the out-dated depth information. This result underscores the importance of **our proposed iterative refinement scheme** for achieving both geometric accuracy and high-quality novel view synthesis.
>
> ### **4\. Q4: Could you provide visualizations of the estimated depth and rendered depth at different stages to demonstrate the effect of online updates?**
> This is an excellent suggestion. Visualizing the depth evolution process provides a clear and intuitive demonstration of our method's core mechanism. Due to the rebuttal format restrictions, we cannot include new figures here. However, the method figure in our main manuscript (Figure 2\) already provides a snapshot of a single update step, showing the rendered depth from the Gaussians (D.E. input) and the refined depth map (D.E. output). This figure illustrates how our method corrects geometric inconsistencies like floating artifacts in a single step. We will add a new, dedicated figure in the final manuscript to show this process over multiple training stages. This visualization will clearly demonstrate how the depth maps are progressively refined from an initial noisy state to a coherent and accurate representation, powerfully illustrating the online update process.

---

> > ### Comment · Reviewer_e8Wz · 2025-08-04
> > **Questions about initialization**
> >
> > Thank you for your detailed rebuttal and response. Regarding the initialization, I still have some concerns:
> >
> > 1. As you pointed out, the experimental datasets already provide camera poses. This means we don’t need to execute a full SfM pipeline—only triangulation and some bundle adjustment (BA) refinements are required to obtain the initial point cloud.
> > 2. Since the point cloud is generated directly from known camera poses, its scale and poses remain inherently consistent. In fact, the triangulated metric depth may even be more accurate than the depth estimated by the network.
> >
> > Given this, I still fail to see the advantage of depth estimation-based initialization over SfM-based point cloud initialization.

---

> > > ### Author Response · Authors · 2025-08-05
> > >
> > > Thank you for this insightful follow-up. In the former answer, we mainly discuss the difference of problem setting between SfM and DepthSplat-based MVS. However, as you mentioned, with known camera poses, one could bypass the full SfM pipeline and use triangulation and bundle adjustment to generate an initial point cloud. Nonetheless, our choice to adopt a learning-based MVS approach is based on three key considerations:
> > >
> > > **1\. Runtime Efficiency:** Traditional SfM pipelines, such as COLMAP, are powerful but computationally expensive. For example, processing a single Waymo sequence of 150 frames to generate a sparse point cloud can take approximately **two hours**. In contrast, DepthSplat produces a dense geometric prior through a single forward pass, generating a dense point cloud in roughly **three minutes**. This significant reduction in processing time makes the overall training process more efficient and practical for real-world applications.
> > >
> > > **2\. Sparsity vs. Density of the Geometric Prior:** In autonomous driving scenes with sparse camera views (150 views for a 80m\*30m\*20m scene), traditional SfM methods tend to produce even **sparser** point clouds due to challenges of limited view, textureless surface and occlusion regions. Moreover, using dense reconstruction with these methods would incur an even greater computational cost. In contrast, our approach prioritizes **dense point clouds** for Gaussian initialization, which is then followed by **dense depth regularization** in the later training phase.
> > >
> > > **3\. Our Core Contribution:** Our paper's core technical contribution is not the specific initializer, but a framework designed to robustly handle **imperfect geometric prior,** i.e. **sparsity** issue of SfM-based approach or **local inaccuracy** in learning-based MVS. Our joint optimization framework is orthogonal to the choice of initialization method.
> > >
> > > This is further supported by the new experiment we presented in our response to **Reviewer GUm1, Tab. A**, where we initialized our pipeline with **UniDepthV2,** a monocular depth estimator with more accurate metric depth. Our method shows consistent improvements regardless of the initialization method.
> > >
> > > Due to time constraints, we are currently unable to provide a full experiment using an SfM-based initialization. However, we are committed to including this ablation study in the final manuscript to quantitatively analyze this method.
> > >
> > > Thank you again for engaging in this important discussion.

---

> > > > ### Comment · Reviewer_e8Wz · 2025-08-05
> > > >
> > > > Thank you for your prompt reply.
> > > >
> > > > 1. If COLMAP takes two hours to triangulate 150 views, I believe this is unreasonable, and there may be issues with your setup. If the author is not familiar with the traditional SfM pipeline, I don’t think we need to spend much time discussing computational efficiency, as it does not affect my evaluation of this work.
> > > >
> > > > 2. The sparsity and density of geometric priors are good points for determining initialization strategies. The lack of initialization priors in textureless or weakly textured regions may lead to degradation of local geometry.
> > > >
> > > > 3. The second contribution explicitly states that the `progressive pruning strategy` is a key contribution of this paper. However, it is clear that this strategy is specifically designed for dense initialization. Therefore, even if the authors emphasize the orthogonality of initialization schemes, it only applies to different choices of dense initialization—not between sparse and dense initialization.
> > > >
> > > > Overall, I am not opposed to dense initialization, but I believe it is crucial to clearly define the boundary between dense and sparse initialization, as this is key to highlighting the innovations of this work. Only if dense initialization demonstrates significant benefits over sparse initialization can some of the novel contributions in this paper be justified.
> > > >
> > > > I believe the current presentation of this work is already sufficiently positive, though it would be even better if the authors could include additional relevant discussions in the future.

---

> > > > > ### Author Response · Authors · 2025-08-06
> > > > >
> > > > > Thank you for your prompt and insightful reply. We appreciate the opportunity to provide further clarification on the following two points:
> > > > >
> > > > > 1. **The Definition of Dense and Sparse Initialization**
> > > > >
> > > > > From our perspective, the definition of "dense" refer to the dense depth estimation, i.e a pixel-wise prediction for depth. Therefore, a "dense initialization" corresponds to the point cloud derived from these **dense depth maps**, yielding a redundant yet geometric comprehensive point representation. In contrast, "sparse initialization/point cloud" refers to point clouds obtained through sparse sampling of the original scene (e.g. LiDAR point cloud, sparse point cloud from COLMAP),  which may miss fine-grained local geometry in certain regions. Moreover, such sparse point clouds can only provide sparse depth maps for supervision. As shown in Figure 3 of the main paper, our dense depth regularization produces smoother and more complete (hole-free) depth results compared to the sparse depth supervision provided by LiDAR.
> > > > >
> > > > > 2. **Orthogonality of Initialization Scheme**
> > > > >
> > > > > We thank the reviewer for raising an excellent point that the Progressive Pruning strategy is specifically designed for a dense initialization.
> > > > > However, we argue that with minor modifications, our pipeline can be adapted to work with sparse point clouds—for example, by removing the Progressive Pruning module and initializing the depth enhancer using sparse depth maps projected from the sparse point cloud. In this setting, reprojection error will also arise similar to what is encountered with LiDAR-based point clouds. Theoretically, the depth enhancer remains a valuable component for refining the depth pseudo-ground truth; however, this hypothesis requires further validation through future experiments.
> > > > >
> > > > > Thank you again for the valuable discussion. We will include this crucial discussion and a full ablation study of comparing different initialization scheme in our final version.

---

> > > > > > ### Comment · Reviewer_e8Wz · 2025-08-06
> > > > > >
> > > > > > Thank you for your response. I don’t have any additional questions, and I’d be happy to raise the final rating.

---

### Official Review · Reviewer_Cnm4 · 2025-07-02

**Clarity:** 3
**Significance:** 3
**Originality:** 3
**Rating:** 4
**Confidence:** 5

**Summary:**

The paper proposes a LiDAR-free method for urban scene reconstruction based on Gaussian Splatting. It introduces a dense point cloud initialization method through metric depth estimation and progressive pruning, a joint optimization framework leveraging diffusion priors to refine both predicted and rendered depth maps, and a Gaussian model for road regions to encourage planar geometry. Experimental results show that the proposed approach outperforms existing baselines and performs better than using ground-truth LiDAR data, which can be prone to errors due to multi-sensor calibration and re-projection.

**Questions:**

1. How does this paper handle dynamic objects? How is dynamic object decomposed? What kind of gaussian model or nodes are used to represent dynamic objects? Does the paper use bounding box or self-supervised method such as PVG? I think this part is missing from the paper.
2. Following the previous point, I am particularly interested in how the proposed method handles depth estimation for dynamic objects. If the rendered depth from Gaussians contains errors or noise especially in dynamic regions, how robust is the method to such artifacts? This is closely related to the type of dynamic object modeling used in the paper, which is currently not well described.
3. The paper claims that LiDAR depth is not always reliable due to calibration or re-projection issues. Given this, I am not sure whether purely comparing the estimated depth directly against ground-truth LiDAR, as shown in Table 2 and Table 3, is fully justified. Instead, it might be more meaningful to report image-level reconstruction metrics (e.g., PSNR, SSIM, LPIPS) using both LiDAR-based and proposed depth to support the method.
4. Following the last point, can the author show the results of using proposed method w/o depth enhancer but with DepthSplat or adding an ablation table showing image recon/synthesis metrics of removing each proposed components? It would be interesting to see which components contribute more to the image qualities and especially how the proposed depth estimation contributes.

**Ethical Concerns:**

["NO or VERY MINOR ethics concerns only"]

**Final Justification:**

My questions have been adequately addressed, and the authors have provided additional evaluation results. Based on this, I am inclined to raise my score.

**Limitations:**

yes

**Quality:**

3

**Strengths And Weaknesses:**

Strengths:
1. The paper is well written and easy to follow.
2. It addresses an important and timely topic, demonstrating that Gaussian Splatting with estimated metric depths can outperform methods relying on ground-truth LiDAR data. This contribution is especially valuable for scenarios in autonomous driving where only camera data is available.

Weaknesses:
1. While the proposed method is promising, the main weakness of the paper lies in its limited experimental evaluation. Currently, results are only presented on the NOTR dataset. Given the paper’s claim of outperforming methods that rely on potentially inaccurate LiDAR data, it would be more compelling to include evaluations on the nuScenes dataset. To my knowledge, LiDAR calibration in nuScenes is generally less accurate than in the Waymo dataset. Validating the proposed method on nuScenes would strengthen the paper’s contribution and make the results more convincing.

---

> ### Author Rebuttal · Authors · 2025-07-31
>
> ### **Dear Reviewer Cnm4,**
>
> We sincerely thank Reviewer Cnm4 for the thorough review and constructive feedback. In this rebuttal, we provide new experimental results on the nuScenes dataset as you suggested, comprehensive ablation study results, and detailed clarifications on our method for handling dynamic objects.
>
> ### **1\. W1. Validating the proposed method on nuScenes would strengthen the paper’s contribution and make the results more convincing.**
>
> We completely agree with your assessment. Adding a new dataset is essential for validating the generalization of our method.  Testing our method in scenarios with greater LiDAR noise is indeed an excellent way to highlight its effectiveness. However, while such data is common in industry, it is rarely available in public datasets. We thank you for suggesting the nuScenes dataset and for noting its less accurate LiDAR calibration. Accordingly, we have extended our method to nuScenes and compared it against OmniRe-based approaches. The experimental results are summarized below.
>
> Tab. A: Quantitative comparison of our method against OmniRe-based approaches on the nuScenes dataset. We select 8 sequences from the nuScenes dataset (two of which are rainy scenarios) due to the time limitations of the rebuttal phase. We will provide the full results in the revised manuscript.
> | **Methods** | **Image reconstruction** | | | | **Novel view synthesis** | | |
> | :--- | :---: | :---: | :---: | :---: | :---: | :---: | :---: |
> | | **PSNR**$\uparrow$ | **SSIM**$\uparrow$ | **LPIPS**$\downarrow$ | | **PSNR**$\uparrow$ | **SSIM**$\uparrow$ | **LPIPS**$\downarrow$ |
> | OmniRe | 30.12 | 0.866 | 0.117 | | 27.50 | 0.787 | 0.166 |
> | OmniRe + DS | 29.78 | 0.900 | 0.127 | | 26.60 | 0.678 | 0.189 |
> | Ours | 30.63 | 0.890 | 0.156 | | 27.22 | 0.724 | 0.175 |
> |  |
>
> As the results show in Tab. A, our method achieves better performance in both image reconstruction and novel view synthesis compared to the LiDAR-free baseline. Moreover, our method remains comparable to the LiDAR-based OmniRe in reconstruction and novel view synthesis. We attribute this to the fact that while the LiDAR data in nuScenes is considered less accurate than Waymo's, it is a mature and widely-used dataset, and its errors are still relatively minor compared to those from image-based depth estimation methods. Due to the NeurIPS rebuttal limitations, we are unfortunately unable to provide visualizations at this time. We will add the experiments on nuScenes, along with corresponding qualitative and quantitative results to the final version of our manuscript to fully demonstrate our method's performance across different datasets and more challenging scenarios.
>
> ### **2\. Q1. How does this paper handle dynamic objects?**
>
> Thank you for pointing out the lack of clarity regarding our handling of dynamic objects. We will add a dedicated section in the final manuscript to elaborate on these technical details. In response to your specific questions:
>
> * **Dynamic Object Modeling:** Our approach to modeling and decomposing dynamic objects follows our baseline, OmniRe, by utilizing a Scene Graph. Each dynamic object is represented as an independent node within this graph, possessing its own set of Gaussian models (e.g., dynamic or rigid representations).
> * **Depth Estimation for Dynamic Objects:** We acknowledge that initial depth estimates derived from MVS will inevitably contain errors for moving objects, a common challenge in depth estimation. First, since our method uses given camera poses from Waymo, dynamic objects do not affect pose estimation. Depth errors are therefore localized to these dynamic areas, with minimal impact on the static background. Furthermore, during the iterative optimization of the Gaussians and the Depth Enhancer, the depth of dynamic objects is further refined. Specifically, this refinement is driven by both the photometric loss from Gaussian splatting and the accurate geometric supervision provided by our diffusion-based Depth Enhancer. This joint optimization loop is **equally effective for both static and dynamic scene components**, allowing it to effectively correct initial errors and achieve accurate depth prediction
>
> We believe a comprehensive visualization of the depth refinement process would best illustrate the optimization of depth for dynamic objects. As requested by Reviewer e8Wz, we will include such visualizations and a supplementary explanation in the final manuscript to demonstrate the rationale and effectiveness of our method.
>
> ### **3\. Q2. Why purely compare the estimated depth directly against ground-truth LiDAR, as shown in Table 2 and Table 3?**
>
> This is a very insightful and reasonable point. We agree that using LiDAR as ground truth for depth evaluation can introduce errors, especially in real-world applications where LiDAR data may be faulty. However, for the Waymo dataset, we believe use as a reference is justified for two reasons:
>
> 1. The LiDAR data in the Waymo dataset is meticulously calibrated and is widely recognized in the academic community as a high-quality dataset.
> 2. The potential **errors in the** **Waymo LiDAR data are minor compared to the errors produced by image-based depth estimation algorithms**. For instance, in most regions, LiDAR typically produces centimeter-level errors.
>
> Therefore, adopting Waymo's LiDAR as an **"upper-bound”**  to evaluate the relative performance of different LiDAR-free methods is meaningful. Although the Waymo dataset cannot clearly reveal the impact of LiDAR calibration errors on the accuracy of reconstruction, it can still prove the effectiveness of our method in LiDAR-free setting, i.e. our method achieves performance comparable to LiDAR-based approaches and significantly outperforms other LiDAR-free methods.
>
> ### **4\. Q3. Adding an ablation table showing image recon/synthesis metrics of removing each proposed component?**
>
> Thank you for this excellent suggestion. Regarding the results of a DepthSplat baseline without our Depth Enhancer, they are presented in Tab. 1 and Tab. 2 of the original manuscript. Additionally, we fully agree that providing image-level metrics in our ablation study is important for evaluating scene reconstruction and novel view synthesis quality. Our original ablation study emphasized geometric metrics because the ill-posed nature of sparse-view reconstruction can be misleading, and models can easily overfit to training views, yielding high PSNR scores that don't reflect true geometric accuracy. We now have updated our main ablation study to include PSNR, SSIM, and LPIPS, and will add the following table to our manuscript.
>
> Tab. B: Image-level performance of all ablation study results in our original manuscript.
> | **R.N.** | **P.P.** | **D.E.** | **$L_{\text{ref}}$** | **$L_w$** | **$L_{\text{smooth}}$** | **Update** | **Image reconstruction** | | | **Novel view synthesis** | | |
> | :---: | :---: | :---: | :---: | :---: | :---: | :---: | :---: | :---: | :---: | :---: | :---: | :---: |
> | | | | | | | | **PSNR**$\uparrow$ | **SSIM**$\uparrow$ | **LPIPS**$\downarrow$ | **PSNR**$\uparrow$ | **SSIM**$\uparrow$ | **LPIPS**$\downarrow$ |
> | × | ✓ | ✓ | ✓ | ✓ | ✓ | Real-time | 32.67 | 0.939 | 0.087 | 27.67 | 0.854 | 0.139 |
> | ✓ | × | ✓ | ✓ | ✓ | ✓ | Real-time | 32.25 | 0.932 | 0.099 | 27.17 | 0.847 | 0.149 |
> | ✓ | ✓ | × | × | × | × | Real-time | 32.77 | 0.939 | 0.087 | 27.52 | 0.851 | 0.143 |
> | ✓ | ✓ | ✓ | ✓ | × | × | Real-time | 32.57 | 0.940 | 0.086 | 27.29 | 0.852 | 0.140 |
> | ✓ | ✓ | ✓ | ✓ | ✓ | ✓ | Periodic | 33.95 | 0.951 | 0.075 | 26.90 | 0.823 | 0.128 |
> | ✓ | ✓ | ✓ | ✓ | ✓ | ✓ | Real-time | 33.63 | 0.947 | 0.079 | 28.04 | 0.863 | 0.130 |
> |  |
>
> As shown in Tab. B, the results indicate our full method achieves the best performance compared to variants where other components are removed, underscoring the effectiveness of our proposed modules. When combined with Table 3 in the main manuscript, the results show that, in most cases, poorer depth quality leads to inferior NVS performance. Notably, the 'Periodic Update' variant shows a significant drop in NVS performance compared to our full model, indicating it has overfitted to the training views' photometry with the out-dated depth information. This result underscores the importance of our proposed iterative refinement scheme for achieving both geometric accuracy and high-quality novel view synthesis.

---

> > ### Comment · Reviewer_Cnm4 · 2025-08-05
> >
> > Thank you for the detailed rebuttal. Overall, my concerns have been well addressed, and the additional results on the NuScenes dataset are provided. I believe the proposed method presents an interesting contribution to the field.

---

> > > ### Author Response · Authors · 2025-08-06
> > >
> > > Thank you very much for your kind and constructive feedback. We're glad the additional results and clarifications addressed your concerns. We truly appreciate your recognition of our work and its contribution to the field.

---

### Official Review · Reviewer_eDMK · 2025-07-03

**Clarity:** 4
**Significance:** 3
**Originality:** 3
**Rating:** 5
**Confidence:** 4

**Summary:**

The submission presents a novel framework for urban scene reconstruction that does not require LiDAR data. Gaussian Splatting are a very active area of research and are often considered when thinking of scene reconstruction in autonomous driving scenarios. One of the limitations of these methods is that they require LiDAR data as well. However LiDAR data might be unavailable, not well synchronized with camera data, or the sensors' extrinsics might be decalibrated. The authors address this issue by proposing a camera-only method based on depth estimation.
Authors propose a method that uses multi-view depth estimation to generate a dense point cloud, then optimized with a pruning strategy proposed by the authors. The depth predictions and Gaussian representations are the refined using a diffusion-based module. The method also uses a road node in the scene graph to improve the accuracy of ground geometry by adding constraints. The method is evaluated on the Waymo dataset and shows improvements in image reconstruction and depth estimation compared to LiDAR-based methods.

**Questions:**

How do you see your method performing on other datasets than the one used? Especially with noisier data or more difficult weather conditions ?

The pruning strategy is presented as a solution to lower the computational cost. The strategy seems effective. Could you provide more details on the required resources and on how the computational complexity scales based on the size and complexity of the scene ?

You point LiDAR-camera calibration as an issue because of potential pose errors, but your method requires still accurate camera pose knowledge. Do you think this might be an issue? Do you have ideas to mitigate the impact of pose estimation on the reconstruction result?

**Ethical Concerns:**

["NO or VERY MINOR ethics concerns only"]

**Final Justification:**

This work can be a strong contribution to the field. Authors answered attentively to the reviewers' questions and made significant efforts. Since I was already convinced, I don't feel the need to update my rating and will leave it at 5.

**Limitations:**

yes

**Quality:**

4

**Strengths And Weaknesses:**

## Strengths
Gaussian Splatting are a very active area of research and promising method for scene reconstruction in the context of autonomous driving. However, existing methods require LiDAR data (which causes issues with spatial and temporal calibration to the camera). Proposing a LiDAR-free approach is thus interesting an impactful, as it could ease real-world use of these methods, and improve their robustness (authors show the low quality results of existing methods when LiDAR issues occur). Several innovative components are proposed, which together create a robust solution. Results on the Waymo dataset show improvements over the state of the art in image reconstruction and depth estimation, despite not using LiDAR. Experiments, quantitative and qualitative evaluations support the work, and results are compared with methods from the state of the art. Visualizations like Figure 3 help to better understand the significance of the improvements. The additional implementation details as supplementary material are also appreciated.

## Weaknesses
This method still requires camera poses, which are not always available or of good quality (similar issue as the camera-LiDAR calibration mentioned by the authors). The code is not made available. While using multiple modalities (such as LiDAR) can be an issue because of calibration, synchronization, etc., sensor fusion is used very frequently in autonomous driving, and these problems can be mitigated, so I am not entirely convinced of the need of using cameras only, but I am however convinced of the interest of proposing such method.

---

> ### Author Rebuttal · Authors · 2025-07-31
>
> ### **Dear Reviewer eDMK,**
>
> We sincerely thank Reviewer eDMK for the insightful review and valuable feedback. In this rebuttal, we provide new experimental results on the nuScenes dataset, a detailed computational analysis of our pruning method, and a discussion on addressing the camera pose dependency.
>
> ### **1\. Q1. Performance of your method on other datasets, especially with noisier data or more difficult weather conditions.**
>
> Thank you for this question. We agree that evaluating on a new dataset is necessary to verify the generalization of our method, and testing on a dataset with noisier LiDAR data would better highlight our method's effectiveness. Regarding a **"noisier dataset",** another reviewer (Cnm4) suggested we test on the nuScenes dataset. We believe this also aligns perfectly with your suggestion. Consequently, we have extended our method to nuScenes and compared it with OmniRe-based approaches. The results are presented in the table below.
>
> Tab. A: Quantitative comparison of our method against OmniRe-based approaches on the nuScenes dataset. We select 8 sequences from the nuScenes dataset (**two of which are rainy scenarios**) due to the time limitations of the rebuttal phase. We will provide the full results in the revised manuscript.
> | **Methods** | **Image reconstruction** | | | | **Novel view synthesis** | | |
> | :--- | :---: | :---: | :---: | :---: | :---: | :---: | :---: |
> | | **PSNR**$\uparrow$ | **SSIM**$\uparrow$ | **LPIPS**$\downarrow$ | | **PSNR**$\uparrow$ | **SSIM**$\uparrow$ | **LPIPS**$\downarrow$ |
> | OmniRe | 30.12 | 0.866 | 0.117 | | 27.50 | 0.787 | 0.166 |
> | OmniRe + DS | 29.78 | 0.900 | 0.127 | | 26.60 | 0.678 | 0.189 |
> | Ours | 30.63 | 0.890 | 0.156 | | 27.22 | 0.724 | 0.175 |
> | |
>
> As the results show in Tab. A, our method achieves better performance in both image reconstruction and novel view synthesis compared to the LiDAR-free baseline. Moreover, our method remains comparable to the LiDAR-based OmniRe in reconstruction and novel view synthesis. We attribute this to the fact that while the LiDAR data in nuScenes is not as high-quality as that in Waymo's, it remains a mature and widely-used dataset, and the discrepancy between its point clouds and the ture depth is still relatively minor compared to the errors in depth estimation. Due to the NeurIPS rebuttal limitations, we are unfortunately unable to provide visualizations at this time. We will add the experiments on nuScenes, along with corresponding qualitative and quantitative results to the final version of our manuscript to fully demonstrate our method's performance across different datasets and more challenging scenarios.
>
> ### **2\. Q2. Detailed computational complexity regarding our Progressive Pruning method.**
>
> Thank you for this question and for recognizing the effectiveness of our Progressive Pruning component. The fundamental reason we proposed this strategy is that initializing from a dense depth map would involve an enormous number of back-projected points (approx. $cams×timestep×H×W$). Using these points directly for Gaussian training would be computationally redundant. Furthermore, an excess of low-accuracy points would lead to artifacts during subsequent training. Therefore, Progressive Pruning is not only a method for optimizing computation but also for providing a higher-quality initial point cloud to improve reconstruction results. Regarding your specific question on computational efficiency, we have profiled the resource usage with different numbers of retained points.
>
>
> **Table B:** Analysis of computational cost, comparing our Progressive Pruning method with a baseline using a manageable number of points to demonstrate the efficiency gains.
> | Method   | Initial Point num.   | Training Point num.* (first 30k iter) | GPU RAM | Runtimes |
> |:--------:|:--------------------:|:-----------------------------------:|:-------:|:--------:|
> | w/o P.P. | $1.5×10^7$                | $6×10^6$ - $8×10^6$                             | 34G     | 10h      |
> | P.P      | $2×10^6$                | $2×10^6$ - $3×10^6$                           | 20G     | 8h       |
> | |
>
> As shown in Tab.B, when the initial input point cloud contains as many as $1.5×10^7$ points, the gaussian model will maintain around $6×10^6$ points that cannot be effectively pruned. In contrast, our Progressive Pruning method prevents a significant waste of computational resources and avoids efficiency bottlenecks.
>
> ### **3\. Q3. Do you think still requiring accurate camera pose knowledge might be an issue? Do you have ideas to mitigate the impact of pose estimation on the reconstruction result?**
>
> Thank you for raising this point; we consider it to be very important. We acknowledge that accurate camera calibration is also a practical challenge in the autonomous driving scenarios. However, our method relies on accurate camera poses, which is a common assumption in the field of street view reconstruction and also a prior knowledge in existing datasets. Therefore,  in this work, we **mainly aim to address problems related to the LiDAR data modality**, such as 1\) the absence of LiDAR data on vision-only autonomous vehicles, 2\) projection errors from different mounting views, and 3\) the inaccuracy of LiDAR calibration.
>
> Regarding how to mitigate the need for accurate camera poses, we have the following thoughts for future work. In the absence of precise poses, we could integrate pose estimation algorithms, such as learning-based methods (e.g., VGGT) or traditional SfM methods (e.g., COLMAP), to obtain initial camera poses. Subsequently, a joint optimization module for camera poses could be used during reconstruction, enabling simultaneous refinement of both camera poses and the reconstructed scene. We will add this discussion to the final version of our manuscript to more comprehensively explain the practical implications of our work and future directions.
>
> ### **4\. W1. The code is not made available.**
>
> Finally, regarding your comment on code availability, we commit to making our code publicly available upon acceptance of the paper.
>
> Thank you again for your thorough review and constructive comments. They have been invaluable in helping us improve the quality and completeness of our manuscript.

---

> > ### Comment · Reviewer_eDMK · 2025-08-05
> >
> > The authors have provided a comprehensive rebuttal that effectively addresses the concerns raised in the review. The additional experiments on the nuScenes dataset demonstrate the method's generalization and effectiveness, particularly in challenging scenarios. The detailed computational analysis of the Progressive Pruning method highlights its efficiency and necessity for high-quality reconstruction. The discussion on camera pose dependency and potential future work to mitigate this issue is insightful and shows a clear path for further improvement. The commitment to making the code publicly available upon acceptance is commendable. Overall, the paper presents a significant contribution to the field of urban scene reconstruction.
> >
> > Authors have provided a very strong and clear rebuttal and addressed the concerns raised in the review. Additional experiments on nuScenes are appreciated and show the method's generalisation. The authors proposed a detailed analysis of the Progressive Pruning method.They confirmed their intention of publishing their code on GitHub upon acceptance.
> > Overall, the paper presents a significant contribution to the field.

---

> > > ### Author Response · Authors · 2025-08-06
> > >
> > > Thank you very much for your encouraging and thoughtful response. We truly appreciate your recognition of our additional experiments, analysis, and we’re glad our clarifications addressed your concerns.

---

### Note · Authors · 2025-08-13

We would like to extend our sincerest gratitude to all reviewers for their insightful feedback.
We are pleased that the reviewers generally recognized the **importance of our work in addressing driving scene reconstruction in a LiDAR-free setting**. Also, **the novelty and effectiveness of our proposed components** was acknowledged as a key strength.
The insightful questions raised during the discussion prompted us to provide further evidences and clarifications, which mainly focus on **the comprehensiveness of the experiments and the motivation behind the initialization method**. We are encouraged and grateful that reviewers found our additional evidence and explanations successfully addressed their concerns. We are confident that these enhancements will make our work significantly more solid and complete, and we plan to add these additional experiments and discussions to the next version.

---

### Decision · Program_Chairs · 2025-09-17

**Decision:**

Accept (poster)

**Comment:**

The paper received (A, BA, A, BA) with confidence (4, 5, 3, 5).
All four reviewers agree that this paper proposes a timely and impactful contribution to LiDAR-free urban scene reconstruction with Gaussian Splatting. The technical innovations, extensive experiments, and practical significance justify publication. Recommended for acceptance.